# ShapeCraft: LLM Agents for Structured, Textured and Interactive 3D Modeling

**Shuyuan Zhang**[1*] **Chenhan Jiang**[2*†] **Zuoou Li**[1] **Jiankang Deng**[1†]

[*] Equal Contribution [1]Imperial College London
[2]Hong Kong University of Science and Technology

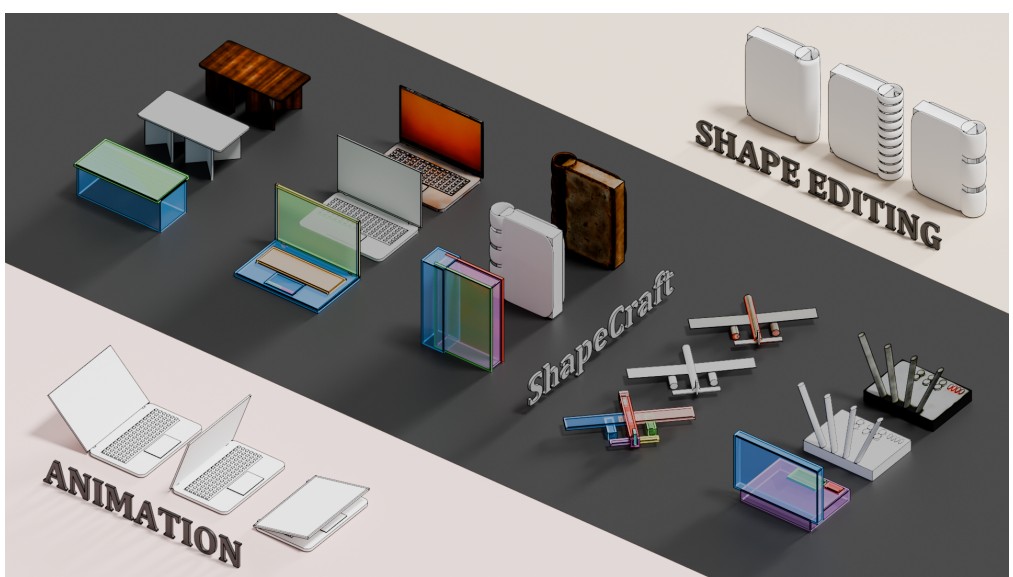

Figure 1: **Qualitative results of ShapeCraft.** Our agentic text-to-shape framework generates bounding volumes, raw meshes and textured shapes, enabling advanced post-modeling interactions like shape editing and animation tasks. Project page is https://sanbingyouyong.github.io/shapecraft.

## Abstract

3D generation from natural language offers significant potential to reduce expert manual modeling efforts and enhance accessibility to 3D assets. However, existing methods often yield unstructured meshes and exhibit poor interactivity, making them impractical for artistic workflows. To address these limitations, we represent 3D assets as shape programs and introduce ShapeCraft, a novel multi-agent framework for text-to-3D generation. At its core, we propose a Graph-based Procedural Shape (GPS) representation that decomposes complex natural language into a structured graph of sub-tasks, thereby facilitating accurate LLM comprehension and interpretation of spatial relationships and semantic shape details. Specifically, LLM agents hierarchically parse user input to initialize GPS, then iteratively refine procedural modeling and painting to produce structured, textured, and interactive 3D assets. Qualitative and quantitative experiments demonstrate ShapeCraft's superior performance in generating geometrically accurate and semantically rich 3D assets compared to existing LLM-based agents. We further show the versatility of ShapeCraft through examples of animated and user-customized editing, highlighting its potential for broader interactive applications.

[†]Corresponding authors: jchcyan@gmail.com, j.deng16@imperial.ac.uk

39th Conference on Neural Information Processing Systems (NeurIPS 2025).

# 1 Introduction

3D modeling plays a pivotal role in domains ranging from immersive entertainment to embodied AI systems. While conventional workflows rely on professional artists using domain-specific tools like Blender [4] or Maya [2], this process is both time-consuming and costly. Recent advances have explored generative methods to democratize 3D content creation through natural language interfaces, yet significant challenges persist in producing production-ready assets.

Current text-to-3D generation systems primarily follow two paradigms. Optimization-based methods [43, 7, 33, 51, 25] leverage pre-trained 2D diffusion models [49] to create implicit 3D representations like neural fields [39] and signed distance field [42]. These require subsequent iso-surfacing [37, 13, 32] to extract usable meshes, often resulting in dense tessellation, smoothing artifacts, and topological inconsistencies [20]. Alternatively, autoregressive approaches [8, 57] directly generate surface meshes by modeling triangle sequences, often training from scratch on large-scale datasets [55, 6, 12]. However, these methods frequently lack semantic part segmentation and exhibit poor modifiability due to their monolithic representation. Both paradigms thus struggle to yield structured and highly editable 3D models for practical artistic workflows. To meet practical demands, an ideal generative modeling system should demand three essential capabilities: 1) production of well-structured geometry with plausible topology compatible with industry workflows; 2) use support for post-modeling interaction, allowing shapes to be easily edited, animated, or argricult; and 3) comprehension of complex, lengthy natural language descriptions.

One promising approach to enable structured and interactive 3D shape generation is to represent shapes as structured computer programs. Such procedural representations [10, 67, 68] not only produce geometry upon execution, but also allow users with basic programming knowledge to understand and modify the generated models [27]. However, conventional methods for shape program generation training on point clouds [3] or CAD modeling datasets [60, 59]. The more general task of text–to–shape program generation remains largely underexplored, mainly due to the scarcity of annotated text–program pairs. Recently, LLM agents [9, 61, 16] have shown potential for translating natural language into programs [26], leveraging their remarkable reasoning and understanding. While this has inspired exploration into using them for shape program generation, a major challenge arises from LLMs' limited ability to interpret complex textual inputs concerning spatial relationships and semantic shape details. Indeed, 3D-PREMISE [66], an initial effort to integrate LLMs with 3D modeling software for direct shape generation, often yield inaccurate programs. CADCodeVerify [1] attempts to improve this with a visual question answering-based self-correction strategy, but it's restricted to CAD datasets and does not generalize to open-domain 3D shapes.

To this end, we introduce a graph-based procedural shape (GPS) representation that breaks down complex natural language descriptions into a structured graph of sub-tasks, augmented with coarse bounding volumes to define their spatial relationships. This decomposition into simpler, inherently independent components significantly enhances the LLM's capacity to understand user descriptions, serving as a shared memory in our generative system. Building upon GPS representation, we design a multi-agent system ShapeCraft, comprising a `Parser`, `Coder`, and `Evaluator`. The `Parser` agent is responsible for constructing the GPS from the initial user input. Subsequently, for the modeling phase, nodes in GPS enable `Coder` agent to employ a multi-path sampling strategy, exploring alternative modeling sequences in parallel while collaborating with the `Evaluator` for validation and refinement. We further develop a component-aware BRDF-based shape painting module for surface appearance. Leveraging the component decomposition provided by GPS nodes, this module improves text alignment for user descriptions and enables realistic surface–light interactions in downstream tools. Qualitative results are showcased in Fig. 1. Extensive experimental evaluation demonstrates that our ShapeCraft can produce more accurate shapes following user input than existing LLM-based methods. Compared to optimization-based and autoregressive 3D generation approaches, ShapeCraft possess superior geometric structure in quality and quantitative results.

Our contributions are summarized as follows:

- Exploration of a graph-based procedural shape representation, facilitating efficient programmatic updates and flexible structure interaction for real-world applications.

- Introducing ShapeCraft, a multiple LLM agents system designed for 3D shape modeling and painting. Our approach leverages the innate multimodal reasoning capabilities of LLMs, streamlining the efficiency of end-users engaged in procedural 3D modeling.

- Empirical experiments demonstrate the substantial potential of LLMs in terms of their reasoning, planning, and tool-using capabilities in 3D content generation.

## 2 Related works

**Text-to-3D Generation.** Existing text-to-3D generation methods can be categorized into two streams: optimization-based methods and autogressive-based methods. The former approaches [43, 7, 33, 58, 25, 23] center around the Score Distillation Sampling (SDS) algorithm proposed by [43], which leverages 2D diffusion model priors[49] for optimizing unstructured 3D representations [39, 30]. However, These require subsequent iso-surfacing [37, 13, 32] to extract usable meshes, often resulting in dense tessellation, smoothing artifacts, and topological inconsistencies [20].The latter one use auto-gressive architectures [5, 28] to directly encode the sequence of triangles. These models are efficient in inference but often struggle with generalizability and training stability, attributed to the limited scope and complexity of available 3D datasets. One promising way is procedural modeling [27] to produce structure 3D shape. However, there is no sufficient text-program pairs for more general classes. In this work, we propose to use the understanding and reasoning capacity of LLM [40] to generate python API for industrial tools.

**LLM Agents.** Recent advances in large language models (LLMs), such as LLaMA [54, 17] and GPT-4 [41], have expanded their capabilities beyond natural language processing to multimodal tasks, particularly in understanding and generation of vision languages. The emergence of LLM-based agents, as seen in works like AutoGPT [18], HuggingGPT [50], and InternChat [36], has demonstrated their ability to autonomously plan and execute complex workflows leveraging external tools, ranging from software development [44, 48, 38] to image synthesis [56, 45]. In image generation, LLM agents have been applied to layout planning [15, 46], self-correction [56], and dynamic model selection [45], significantly improving controllability and adaptability in text-to-image pipelines.

However, compared to their sophisticated applications in 2D counterparts, the adoption of LLM agents in 3D remains relatively limited. Existing efforts primarily focus on scene generation[63, 22, 34, 35, 68]. These methods demonstrate the potential of LLM agents for spatial understanding, translating natural language descriptions into layouts [63] or scene graphs [22, 34]. They also highlight robust tool integration, as 3D-GPT [53] models 3D scenes via function calls to an existing procedural function library. Nevertheless, these approaches predominantly retrieve and arrange existing 3D assets to populate scenes, capturing only coarse inter-object spatial relationships. Critically, they lack direct support for fine-grained shape modeling, which demands a more complex semantic understanding and precise geometric detailing beyond simple layout generation or asset placement.

Preliminary efforts in shape modeling [66, 62] often suffer from the LLMs' limited ability to produce geometrically sound structures. Despite enhancements like CADCodeVerify's [1] VQA-based feedback and BlenderLLM's [14] fine-tuning on specific instructive prompts and shape pairs, these methods primarily cater to specialized CAD modeling tasks. This specialization hinders their generalization to open-ended natural language prompts, which inherently involve greater geometric and semantic complexity. To bridge the gap, we propose a multi-agent system equipped with a novel graph-based procedural shape representation supporting more diverse and complex shape generation.

## 3 Method

### 3.1 Overview of ShapeCraft

The proposed ShapeCraft is a collaborative multi-agent system designed to tackle the complex text-to-3D generation task, as depicted in Figure 2. The system's architecture features three specialized agents—a Parser, a Coder, and an Evaluator—that interact by a central, shared data structure: the Graph-based Procedural Shape (GPS) representation. Each agent has a distinct role in progressively constructing and refining this shared representation:

`Parser` agent is responsible for establishing the initial topology of the GPS representation by parsing the input text into the graph's nodes, edges, and their associated semantic descriptions (in Sec. 3.2).

`Coder` agent populates the GPS representation with concrete attributes. For each node, it can generate the corresponding bounding volume or code snippet that defines its geometry (in Sec. 3.2 and 3.3).

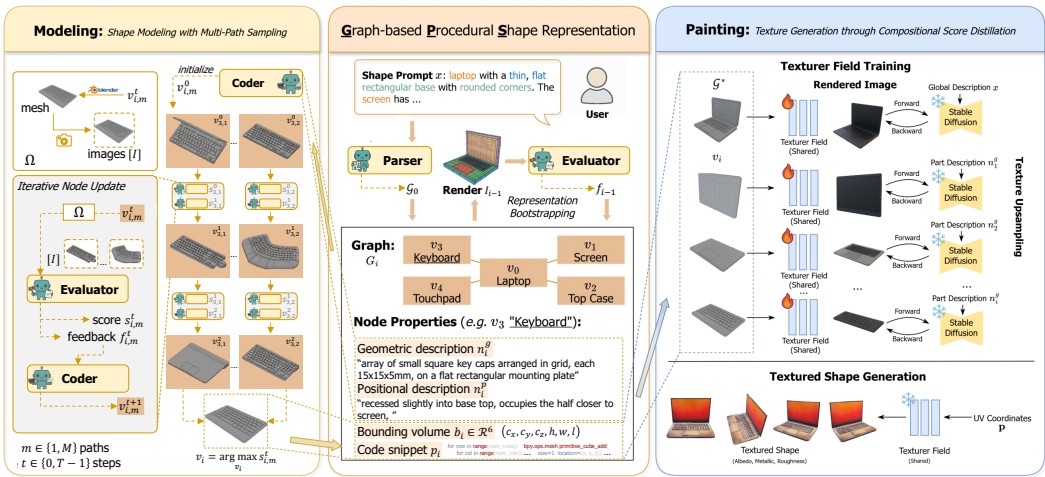

Figure 2: **Overview of ShapeCraft system,** a multi-agent system to produce structured and post-modeling friendly 3d assets. Given a shape description $x$, the `Parser` agent hierarchically decomposes the shape and initializes **G**raph-based **P**rocedural **S**hape representation $\mathcal{G}$. Then, each node $v_i$ is iteratively modeled by updating its code snippet using a multi-path strategy, with reinforcement from the `Coder` and `Evaluator` agents. Finally, a component-aware score distillation learns a texture field from the resulting mesh to produce textured results.

`Evaluator` agent acts as a quality control mechanism. It assesses the outputs generated by the Coder, providing feedback on the plausibility of the bounding volumes and the correctness of the code snippets to guide the self-correction process (in Sec. 3.2 and 3.3).

Once the GPS representation is updated or finalized, a procedural execution module $\Omega$ is invoked. This module executes the code snippets stored within the GPS for each node in Blender. It then assembles the geometry of the resulting primitives based on respective bounding volumes, forming the complete 3D shape. If a code snippet for a node is empty, the module defaults to generating a primitive cube parameterized by that node's bounding volume. When given a specific node of GPS, $\Omega$ bypasses the full assembly process and directly yields the corresponding partial geometry.

## 3.2 Graph-based Procedural Shape Representation

Central to our agentic framework is a GPS representation $\mathcal{G} = (\mathcal{V}, \mathcal{E}, \mathcal{A})$, serving as a shared memory for all agents. Although complex shape parsing follows a hierarchical breakdown, the GPS representation employs a flat, single-level graph structure. This design facilitates parallel shape modeling by treating each geometric component as a self-contained task. The graph is rooted in a semantic virtual root node, $v_0 \in \mathcal{V}$, representing the global abstraction. All other nodes, $\{v_i\}_{i>0} \subset \mathcal{V}$, represent distinct geometric components, and are treated as direct children of the root. This results in a depth-1 structure where the edges are defined as $\mathcal{E} = \{(v_i, v_0)|i > 0\}$. Each component node $v_i$ is then characterized by four attributes $\mathcal{A}(v_i) = (n_i^g, n_i^p, b_i, p_i)$, where:

- Geometric description $n_i^g$: A upsampled textual description of component $v_i$, emphasizing its specific geometric shape and features. By narrowing the LLM's attention to a component-level searching space, this enhances the accuracy of code generation.

- Positional description $n_i^p$: A textual description outlining the spatial relationships and relative placement of $v_i$, which guides the LLM in determining its bounding volume parameters.

- Bounding volume $b_i \in \mathbb{R}^6$: Defines the spatial extent of $v_i$ with its center coordinates and size $(c_x, c_y, c_z, h, w, l)$. This geometric information is crucial for accurately positioning the component and normalizing its scale, ensuring overall consistency in the complete 3D shape.

- Code snippet $p_i$: An executable Blender API script, initially empty and ensuring accessibility and comprehensibility for LLMs.

**Hierarchical shape parsing and graph initialization.** Given the input $x$, `Parser` first instantiates the virtual root node $v_0$, summarizing the core identity of the described object. Subsequently, the

agent iteratively decomposes this high-level concept into a conceptual hierarchy of components. Crucially, this hierarchy is then flattened to conform to our GPS representation. Only the terminal nodes of these decomposition paths are retained to form the set of component nodes $\{v_i\}_{i>0}$. For example, after summarizing $x$ as "chair", the `Parser` might generate a reasoning path such as "chair→upper body→backrest". And "backrest" is instantiated as a node $v_i$, which is then connected directly to the root $v_0$.

Following hierarchical parsing and structural flattening, which establishes the graph's topology $(\mathcal{V}, \mathcal{E})$, `Parser` agent further generates geometric and positional descriptions $n_i^g$ and $n_i^p$ respectively. Subsequently, the Coder agent utilizes positional description $n_i^p$ to generate a corresponding bounding volume $b_i$ for each component, thus completing the initialization of the skeleton graph1 $\mathcal{G}_0$.

**Representation bootstrapping.** To mitigate potential inaccuracies in the GPS representation arising from the inherent limitations or hallucinations of LLMs, we propose a representation boot-strapping process. We aim to enhance an initial representation $\mathcal{G}_0$ to produce a more accurate final version $\mathcal{G}^*$. As mentioned above, initial representation $\mathcal{G}_0 \leftarrow \texttt{Coder}(\texttt{Parser}(x))$. Then for each iteration $i$, the following two steps are performed:

1. **Evaluation and feedback generation:** The current representation $\mathcal{G}_i$ is assessed by an `Evaluator` agent, which inspects the rendered bounding box images for components $v_0 \in \mathcal{G}_i$ to identify inconsistencies or errors. It then produces a textual feedback $f_i$, outlining the necessary corrections: $f_i = \texttt{Evaluator}(\Omega(\mathcal{G}_i))$

2. **Conditional graph update:** The original description $x$, the feedback $f_i$, and the last $\mathcal{G}_i$ are used as a combined context to generate an improved representation $\mathcal{G}_{i+1} \leftarrow \texttt{Coder}(\texttt{Parser}(x, f_i, \mathcal{G}_i))$. In this step, the `Parser` re-interprets the input $x$ conditioned on the previous information, and the `Coder` subsequently refines the bounding box parameters.

The process terminates after $N$ iterations, yielding the final refined representation $\mathcal{G}^* = \mathcal{G}_N$. Empirically, we find $N = 2$ is a good trade-off between performance and computational efficiency.

### 3.3 Iterative Shape modeling with Multi-path Sampling.

To address the `Coder` agent's inherent limitations in spatial understanding and generate diverse and accurate 3D shapes, we propose an iterative shape modeling with multi-path sampling strategy. For capturing 3D modeling diversity and enabling broader exploration of design alternatives, multi-path sampling strategy is employed by configuring the `Coder` agent with higher temperature settings, thereby encouraging the generation of multiple, distinct modeling paths for each shape component. Iterative modeling aims to correct the `Coder` when unreasonable results are produced. The entire process is detailed in Algorithm 1.

Initially, for each node $\{v_i\}_{i>0} \subset \mathcal{V}$ within $\mathcal{G}^*$, we create copies of the node's state, denoted as $\{v_{i,m}^0\}_{m=1}^M$, where $M$ is the number of modeling paths, superscript 0 indicates the initial iteration step. The `Coder` agent then populates each $v_{i,m}^0$ with its corresponding initial code snippet, based on the node's geometric description $n_i^g$ supplemented by a textual overview of $\mathcal{G}^*$.

Subsequently, the multi-path modeling proceeds iteratively for $T$ refinement steps (or until early stopping). At each step $t \in \{0, ..., T-1\}$, `Evaluator` agent provides evaluations for $v_{i,m}^t$, using the procedural execution module $\Omega$ to assemble and render images of the resulting component from different viewing angles (detailed camera settings can be found in Appendix Section B). A feedback description $f_{i,m}^t$ and a quantitative quality score $s_{i,m}^t$ are generated, formally expressed as:

$$f_{i,m}^t, s_{i,m}^t = \texttt{Evaluator}(\Omega(v_{i,m}^t)) \tag{1}$$

Following this evaluation, the `Coder` agent updates and refines these candidate nodes for the next iteration. This self-correction process for each node $v_{i,m}^t$ in a path is driven by its $n_{i,m}^g$ and $p_{i,m}^t$, in conjunction with the `Evaluator`'s feedback and $\mathcal{G}^*$, leading to the updated node:

$$v_{i,m}^{t+1} \leftarrow \texttt{Coder}(v_{i,m}^t, f_{i,m}^t, \mathcal{G}^*) \tag{2}$$

We also employ an early stopping mechanism to improve efficiency: if any candidate path achieves a score $s_{i,m}^t$ higher than a preset threshold $s_\tau$, its generation result is deemed sufficient and the iterative process is terminated to conserve LLM computational resources.

**Algorithm 1:** Iterative Shape Modeling with Multi-path Sampling

---

**Input:** GPS graph $\mathcal{G}^*$, number of paths $M$, maximum iterations $T$, score threshold $s_\tau$
**Output:** Updated GPS representation $\mathcal{G}^*$

1 **for** *each node $v_i \in \mathcal{V}, i > 0$* **do**
2   Create $M$ initial node states: $\{v_{i,m}^0\}_{m=1}^M$
3   Initialize candidate scores: $\{s_{i,m}^{best}\}_{m=1}^M \leftarrow 0$
4   Initialize best node states: $\{v_{i,m}^{best}\}_{m=1}^M \leftarrow \emptyset$
5   **for** *path $m \leftarrow 1$ **to** $M$* **do**
6    Generate initial code snippet from node attributes: $v_{i,m}^0 \leftarrow \texttt{Coder}(\mathcal{A}(v_{i,m}^0), \mathcal{G}^*)$
7    **for** *iteration $t \leftarrow 0$ **to** $T-1$* **do**
8     Execute program: $mesh, [I] \leftarrow \Omega(v_i)$
9     Evaluate: $f_{i,m}^t, s_{i,m}^t \leftarrow \texttt{Evaluator}([I])$
10     **if** $s_{i,m}^t > s_{i,m}^{best}$ **then**
11      $s_{i,m}^{best} \leftarrow s_{i,m}^t$
12      $v_{i,m}^{best} \leftarrow v_{i,m}^t$
13     **if** $s_{i,m}^t \geq s_\tau$ **then**
14      **break** ;         // Early stopping for this path
15     $v_{i,m}^{t+1} \leftarrow \texttt{Coder}(v_{i,m}^t, f_{i,m}^t, \mathcal{G}^*)$ ;      // Refine program
16   $m^* \leftarrow \arg\max_m s_{i,m}^{best}$ ;          // Select best path
17   Update $\mathcal{G}^*$ with $v_{i,m^*}^{best}$
18 **return** $\mathcal{G}^*$

---

Finally, for each node, the candidate path yielding the highest score $s_{i,m}^t$ will be selected. The geometry corresponding to this chosen path will then be used to update the GPS representation $\mathcal{G}^*$, thereby producing the final $\hat{\mathcal{G}}^*$ for subsequent painting or post-modeling interaction.

### 3.4 Component-aware BRDF-based Shape Painting

We propose a component-aware score-distillation sampling scheme, which enables mesh painting from complex prompts by leveraging the compositional structure inherent in GPS representations.

**Texture Field $\psi$.** Let $\mathbf{p} \in \mathbb{R}^2$ denotes UV coordinates on the surface mesh. We define a learnable texture field $\psi_\theta : \mathbb{R}^2 \to \mathbb{R}^5$ mapping UV coordinates to BRDF parameters $(k_d, k_r, k_m) = \psi_\theta(\mathbf{p})$ , where $k_d \in \mathbb{R}^3$ represents the diffuse albedo, $k_r \in \mathbb{R}$ encodes surface roughness measuring the extent of specular reflection, and $k_m \in \mathbb{R}$ denotes the metalness factor. All BRDF parameters are between $[0, 1]$, which can seamlessly integrate into standard rendering pipelines and industrial tools.

**Component-aware Score Distillation (CASD).** We optimize $\theta$ of $\psi_\theta$ by distilling from a pre-trained text-to-image diffusion model through Score Distillation Sampling (SDS) [43] optimization. Given randomly sampled viewing direction $\omega$ and predicted BRDF parameters, a render image $\mathbf{I} = L(\psi_\theta(\mathbf{p}), \omega)$ is obtained following the rendering equation [29]. Then, the parameter $\theta$ is updated by minimizing the SDS loss, whose gradient is computed as:

$$\nabla_\theta \mathcal{L}_{SDS}(\mathbf{I}, x) = \mathrm{E}_{t, \epsilon_\Phi}[w(t)(\epsilon_\Phi(\mathbf{I}_t, t, x) - \epsilon)\frac{\partial g(\theta, c)}{\partial \theta}], \tag{3}$$

where $w(t)$ is weighting function depending on timestep $t$, and $\epsilon_\Phi := (1+s)\epsilon_\Phi(\mathbf{I}_t, t, x) - s\epsilon_\Phi(\mathbf{x}_t, t, \emptyset)$ is the modification of noise prediction with classifier-free guidance (CFG) as $s$.

Our component-aware optimization process integrates both global and component-level SDS to enhance alignment with the user input $x$. The process begins with the constructed GPS representation $\hat{\mathcal{G}}^*$. First, a global UV parameterization $\mathbf{p}$ is computed for the entire shape surface by xatlas [64]. Concurrently, for each component $v_i \in \mathcal{V}$, we isolate the corresponding set of surface points $\mathbf{p}_{v_i}$. Each set exclusively comprises the points on the externally visible surface of its component, enabling

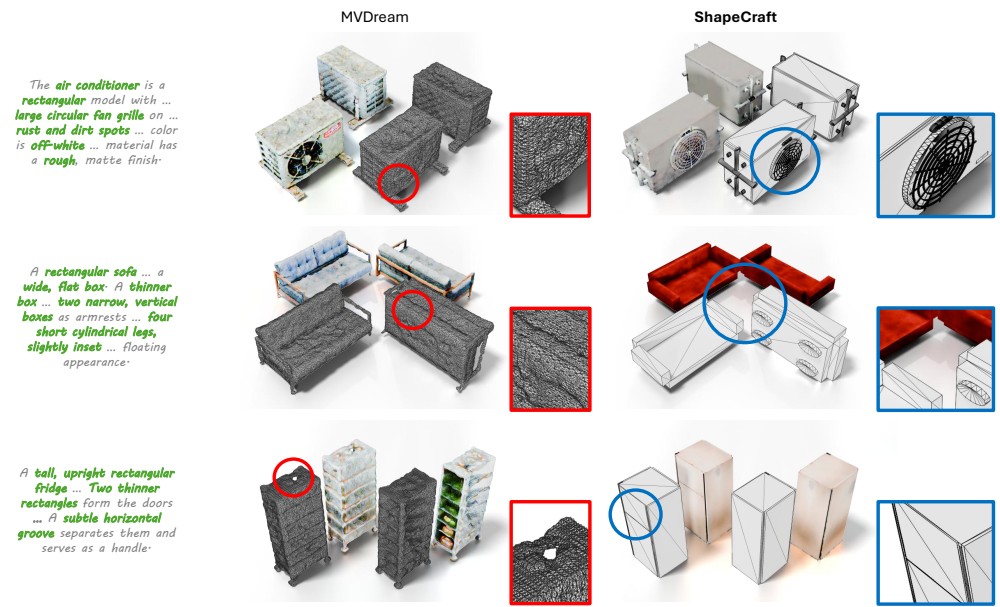

Figure 3: **Qualitative comparison with optimization-based method.** ShapeCraft consistently produces more structured meshes with better prompt following in both geometry and texture (e.g. "rust and dirt spots" in air conditioner). Red and blue areas highlight specific zoom-in observations.

targeted optimization. The final component-aware SDS loss is defined as follow:

$$\mathcal{L}_{CASD} = \mathcal{L}_{SDS}(L(\psi_\theta(\mathbf{p}), \omega), x) + \sum_{i=1}^{M} \mathcal{L}_{SDS}(L(\psi_\theta(\mathbf{p}_{v_i}), \omega), n_i) \quad (4)$$

## 4 Experiments

### 4.1 Implement Details

We employ the same Qwen3-235B-A22B with thinking disabled as `Parser` and `Coder` agents, but the previous one focus on decompose the user input into geometric description and positional description, the latter one transfer geometric description and positional description to bonding box code and shape program. And Qwen-VL-Max as the `Evaluator` agent. For shape modeling, we set the number of path $M = 3$ and the iterative update step $T = 3$ for each node. More experiment settings can be found in Appendix Section B.

**Compared Baselines.** We compare ShapeCraft against a diverse set of baselines, encompassing different paradigms. Our compared methods include the optimization-based method MVDream [51] (assessed with texture), the autoregressive-based method LLaMA-Mesh [57] and several LLM-based methods: 3D-PreMise [66], CADCodeVerify [1], L3GO [62], and BlenderLLM [14]. As the original implementations were not open-source, we reproduced 3D-PreMise, which involves directly querying an LLM to generate an entire shape program and iteratively refining it with visual feedback. Similarly, we reproduced CADCodeVerify, incorporating its Visual Question-Answering (VQA) mechanism to enhance the quality of visual feedback generation. For other methods, we utilized their official codebases with default settings. To ensure a fair comparison, all LLM-based methods that perform a similar function to ShapeCraft's LLM agents (e.g., code generation) employed the same underlying LLM model, specifically Qwen3-235B-A22B, and Qwen-VL-Max for VLM model.

### 4.2 Text-to-Shape Modeling

**Qualitative Comparisons.** **(i) Compared to optimization-based method.** We choose MVDream as a representative optimization-based method. As shown in Figure 3, MVDream struggles to produce structured geometry due to the inherent limitations of extracting surfaces from implicit

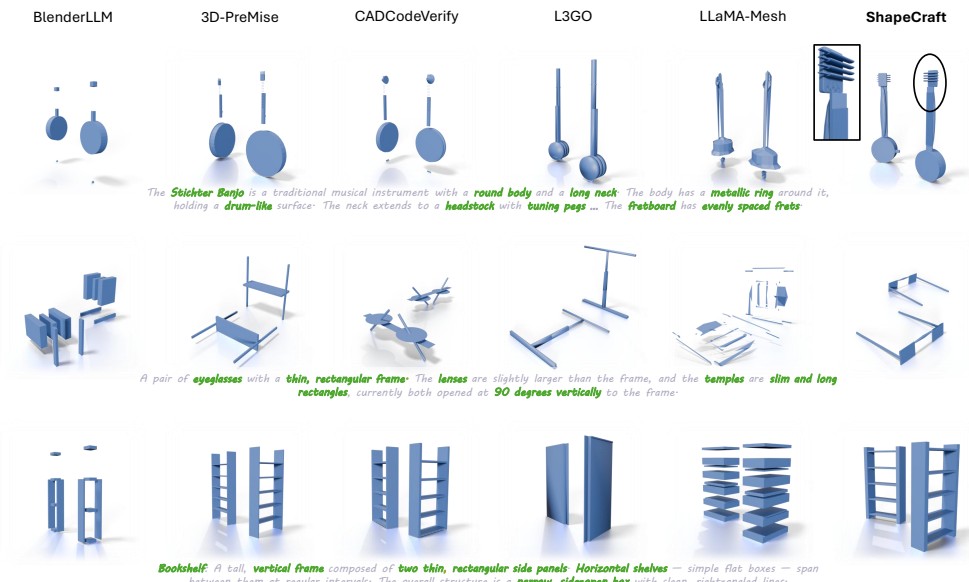

Figure 4: **Qualitative comparison of raw mesh against LLM-based methods.** ShapeCraft demonstrates superior performance for both intricate ("Banjo") and simpler ("bookshelf") cases. The black highlighted areas reveal ShapeCraft's capability to generate complex shape details, benefiting from component decomposition in GPS representation.

Table 1: Quantitative comparison of geometry quality and text-3D consistency on MARVEL subset.

| Methods | IoGT↑ | Hausdorff dist.↓ | CLIP Score↑ | VQA Pass Rate↑ | Run Time↓ | API Calls↓ |
|---|---|---|---|---|---|---|
| 3D-PREMISE [65] | 0.385 | 0.527 | 26.76 | 0.33 | 2.81 min. | 6 |
| CADCodeVertify [1] | 0.334 | 0.511 | 25.94 | 0.34 | 3.06 min. | 9 |
| BlenderLLM [14] | 0.455 | 0.511 | 26.99 | 0.43 | 5.11 min. | N.A |
| LlaMA-Mesh [57] | 0.346 | 0.464 | 25.72 | 0.28 | 15.64 min. | N.A |
| MVDream [51] | 0.427 | **0.411** | 26.84 | 0.42 | 32.10 min. | N.A |
| **ShapeCraft** | **0.471** | 0.415 | **27.27** | **0.44** | 11.68 min. | 21 |

3D representations. This often leads to artifacts such as holes, dense tessellation, and inconsistent topology, as observed in the zoom-in red area. In contrast, ShapeCraft produces accurate topology and smooth surfaces. Furthermore, it demonstrates superior prompt following for both shape modeling and texture generation, as exemplified by "short cylindrical legs, slightly" for sofa case and "dirt spots and rust" for air conditioner. **(ii) Compared to autoagressive and LLM-based methods.** As shown in Figure 4, most methods can deliver acceptable meshes or simpler shapes like "bookshelf", but with increasing difficulty, the mesh generation results for "Stichter Banjo" and "eyeglasses" become inconsistent. We observe that LLM-based approaches have issues identifying or organizing multiple components, or with a significant downgrade in level of details, whereas autoagressive approaches LLaMA-Mesh are limited by training distribution, thus cannot generalize to arbitrary objects.

**Quantitative Comparison.** We conduct evaluations on mesh quality and prompt alignment in Table 1. Our method achieves the best IoGT score, CLIP score and VQA Pass Rate, while also a close second to MVDream in Hausdorff Distance. This suggests ShapeCraft's superiority in terms of prompt following and alignment with shape description, as well as better mesh quality comparing to LLM-based and transformer based methods. Our method is also the only prompting based method that achieves comparable performance to optimization based method with huge advantage in terms of runtime and inference cost.

## 4.3 Ablation Studies

**Ablations on Multi-Path Sampling.** We showcase sampled paths with distinct modeling strategies in the right of Figure 5. Each path is refined with VLM feedback independently increasing robustness and reducing sensitivity to any single LLM failure. We also analyze the diversity of our multi-path

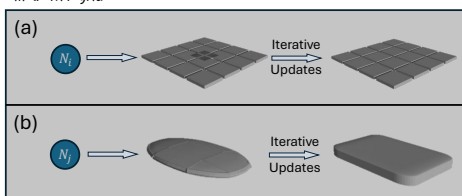

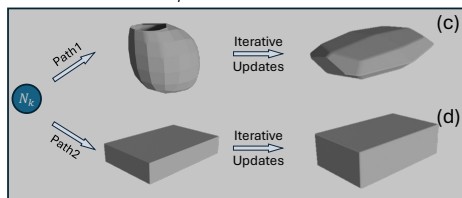

Figure 5: Showcases of iterative refinement for multi-path sampling. (a) Corrects z-fighting artifacts on extraneous buttons. (b) Eliminates redundant bevel operations. (c) Shows a trajectory degraded by a poor initial sample, while (d) demonstrates an alternative path that still yields acceptable geometry.

Table 2: **Ablation studies on sampled paths M and iterative updates T in shape modeling.** Lower Hausdorff and runtime are better, and higher IoGT and CLIP Score are better. ShapeCraft demonstrates a strong balance between exploration and efficiency.

| Metric | M=1, T=1 | M=3, T=1 | M=1, T=3 | **ShapeCraft** (M=3, T=3) | M=3, T=5 |
|---|---|---|---|---|---|
| Hausdorff ↓ | 0.485 | 0.444 | 0.494 | 0.415 | **0.360** |
| IoGT ↑ | 0.436 | **0.535** | 0.492 | 0.471 | 0.431 |
| CLIP Score ↑ | 25.75 | 25.90 | 26.20 | **27.27** | 26.39 |
| Run Time (min) ↓ | **1.62** | 3.71 | 3.90 | 11.68 | 18.04 |

sampling strategy in the shape modeling process. Empirically, we observe that for simple cases, paths tend to converge on similar modeling strategies due to low ambiguity. In contrast, for complex or ambiguous prompts, distinct paths often produce different shape programs, especially under higher temperature settings. We generally observe 2–3 unique strategies across 3 paths with intricate descriptions, reflecting ShapeCraft's capacity to explore diverse modeling alternatives thanks to multi-path sampling. Additionally, we conduct quantitative ablation studies for the first two columns in Table 2. These results show that multi-path sampling significantly improves both geometry quality (IoGT, Hausdorff) and prompt alignment (CLIP), confirming its effectiveness.

**Ablations on iterative updates within path**    We demonstrate the effectiveness of iterative updates guided by VLM feedback within each path on the left of Figure 5. For example, VLM feedback corrects a model clipping issue for calculator buttons and allowed for a more natural modeling of the calculator's case by adjusting the beveling parameters. Additionally, we conduct quantitative ablation studies on sampling configurations, specifically the number of sampled paths $M$ and iterative updates $T$, detailed in Table 2. Increasing the number of sampled paths $M$ consistently shows continuous improvement across all quality metrics. While increasing T does improve the Hausdorff metric, it doesn't guarantee better performance on the other two metrics and introduces greater time overhead. Therefore, ShapeCraft balances performance and efficiency by selecting $M = 3$ and $T = 3$.

Table 3: **Ablation study on hierarchical shape parsing in GPS representation.** We compare with advanced LLMs operating with thinking mode. The results show our GPS representation constrains the reasoning space of LLMs, leading to more reliable and interpretable.

| Metrics | ChatGPT-o3 | ChatGPT-o4-mini-high | Deepseek-R1-0528 | Gemini-2.5-Pro | **ShapeCraft** |
|---|---|---|---|---|---|
| IoGT ↑ | 0.177 | 0.244 | 0.326 | 0.102 | **0.471** |
| Hausdorff ↓ | 0.708 | 0.493 | 0.489 | 0.586 | **0.415** |
| CLIP ↑ | 25.48 | 26.30 | **29.01** | 27.31 | 27.27 |
| Compile Rate ↑ | 60% | 80% | 80% | 60% | **100%** |

**Compared to advanced LLMs with the thinking mode.**    To demonstrate the effectiveness of the `Parser` agent, which performs the explicit hierarchical shape parsing for GPS representation initialization, we compare its performance against advanced LLMs that utilize a thinking mode or Chain-of-Thought (CoT) reasoning. We conduct experiments using the same prompts as in Table 1, enabling the thinking mode for the advanced LLMs, including latest GPT models [24], Deepseek-R1 [19] and Gemini-2.5 [11]. Beyond the metrics for assessing geometry quality, we also report the

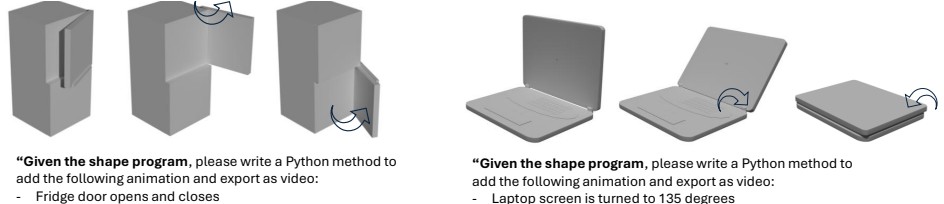

"**Given the shape program**, please write a Python method to add the following animation and export as video:
- Fridge door opens and closes

"**Given the shape program**, please write a Python method to add the following animation and export as video:
- Laptop screen is turned to 135 degrees

Figure 6: **Demonstrating ShapeCraft's flexible post-modeling animation**, where the LLM is prompted to directly generate animation operations based on the existing shape program.

compilation rate. This is defined as the percentage of prompts that successfully produce a valid and compilable 3D shape during a single execution run. As shown in Table 3, we empirically observe that LLMs employing free-form CoT reasoning often struggle to maintain spatial consistency across different components and frequently produce invalid results due to redundant steps or hallucinated geometry operations. In contrast, our ShapeCraft leverages an explicit hierarchical shape parsing mechanism, initiated from a semantic abstract representation. This approach effectively constrains the reasoning space of LLMs, leading to more reliable and interpretable initial shape generations.

**Post-modeling animation.**    The programmable nature of our GPS representation makes it highly amenable to post-modeling interaction, including shape editing and animation. Rather than initially segmenting the holistic object into parts or iteratively optimizing the 3D representation at each timestep, our code snippet for all component nodes can be directly submitted to the LLM and serve as a starting point for further interaction. Figure 6 showcases the seamless export of direct animation from Blender. This is achieved by simply prompting the LLM and provide it with the underlying shape program derived from our GPS representation.

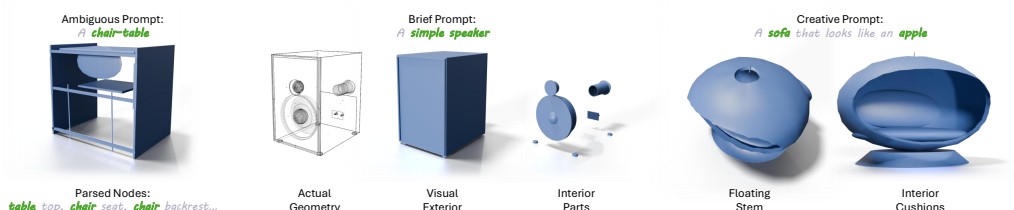

Figure 7: **Failure cases primarily showcase issues arising from ambiguous, brief, and creative prompts.** Ambiguous prompts prevent the `Parser` agent from achieving accurate node decomposition. Brief prompts compromise the `Evaluator` agent's visual signal, leading to invalid iterative updates. Creative prompts confuse the system, often resulting in suboptimal component placement.

## 5   Conclusion

In this work, we introduced ShapeCraft, a multi-agent framework that bridges the gap between text-to-3D generation capabilities and the requirements of practical artistic workflows. Our core innovation is the Graph-based Procedural Shape (GPS) representation, which explicitly converts natural language into a structured task graph. LLM agents within ShapeCraft leverage GPS to hierarchically parse and iteratively refine procedural modeling. Both qualitative and quantitative results demonstrate that ShapeCraft outperforms existing methods and successfully yields structured, textured, and interactive 3D assets, enabling language-centric 3D content creation for artists and developers alike.

**Limitations and failure cases.**    One challenge is the impact of prompt quality on LLM agents, a difficulty that persists even with hierarchical shape parsing and iterative visual feedback. As shown in Figure 7, failure cases primarily arise from three prompt types: ambiguous prompts, which prevent Parser from achieving accurate node decomposition; brief prompts, which compromise the Evaluator's visual signal, leading to invalid updates; and creative prompts, which confuse the system and result in suboptimal component placement. The other main constraint of the current system is the difficulty in producing complex or organic geometry (e.g., tails or wings), a restriction inherent to the Coder agent's library scope. We try to address this by expanding the library to incorporate native 3D models as external components, as shown in Appendix D.

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

## A  Appendix

This supplementary material consists of five parts, including technical details of the experiment and evaluation (Sec. B), additional ablation analysis (Sec. C), additional quality results (Sec. D) and the prompts design (Sec. E).

## B  Technical Details

**Experiment Details.**  Apart from set-up discussed in Section 4, we provide the following additional details: we set a uniform sampling temperature of 0.5 across all LLM and VLM queries, allowing up to three retries in terms of network failure; the visual evaluation score is ranged from 0 to 10 and an early-stopping threshold of 9 is applied; we allow up to one update of the GPS representation $G$ during representation bootstrapping, effectively setting $N = 1$. Beyond shape modeling, bounding volume generation during GPS representation initialization also undergoes an iterative update process by the Coder agent. For this part, we set $M = 1$ and $T = 3$ for its iterative refinement. To provide visual feedback, we render bounding boxes, component shapes and global shapes from 3 preset camera angles - 2 three-quarter views from front-left and front-right, and 1 top-down view from the rear.

**Evaluation Setup.**  All evaluations are performed on the exported meshes. We benchmark on 26 long-form functional prompts from MARVEL-40M+ [52], itself derived from Objaverse [12]. To quantify mesh fidelity, we report Intersection-over-Ground-Truth (IoGT) and Hausdorff Distance (HD) against both sampled point clouds between ground-truth meshes and generated meshes, following [1]. For text–3D alignment, we adopt the CLIP ViT–B/32 [47] as the feature extractor and average the CLIPScore [21] across ten rendered views. We also introduce a VQA-based alignment metric: for each prompt, we author five yes/no/unclear questions, render multi-view images for each method, and compute a VQA pass rate by querying a visual-language model on those questions.

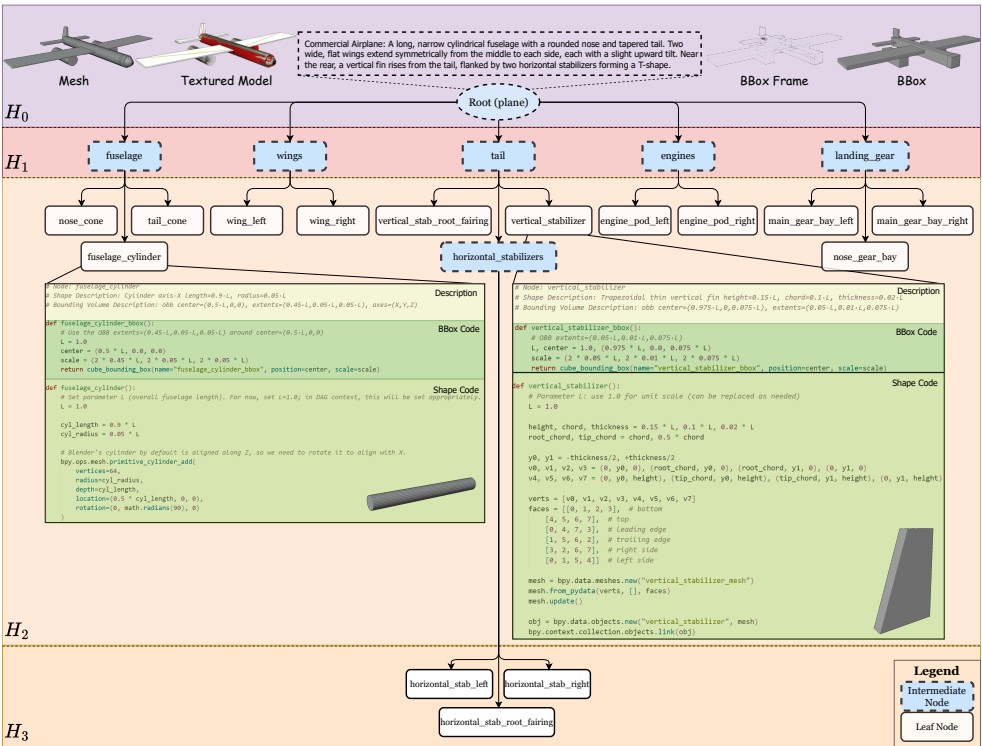

Figure 8: Example of GPS representation for a commercial airplane.

**Wrapped Blender Libraries.**     To constrain the LLM agent's action space, we craft a suite of thin-wrapper methods atop the Blender API that encapsulate frequently used operations, of which the documentation and method signature is illustrated as Prompt 7. The resulting utility library significantly narrows the API surface the LLM must explore.

## C   Additional Ablation Analysis

**Showcase of GPS Representation.**    We illustrate a sample GPS representation for a commercial airplane in Figure 8. The visualization highlights the hierarchical layers and leaf nodes; for selected components, we annotate their geometric parameters, bounding-volume descriptions, and the corresponding shape-generation code (including both modeling and bounding-box routines).

**Bootstrapped Update of GPS representation.**     We show the effectiveness of representation bootstrapping through one example in Figure 9.For the initial graph representation, the agent intend to model two fridge doors together, which is a valid design decision - however, when it tries to set the height of the handle bar, there is no clear indication of where it should go, as the gap between two door panels is not set until the node modeling phases. This may result in mismatched handle bar position and can only be corrected during global optimizations. Our bootstrapped representation spots and solves this issue via a more detailed shape decomposition, modeling two door panels separately and thus the handle bar has a set height independent of subsequent shape modeling processes, guaranteeing a valid shape. At the same time it chooses to model each face of the fridge body individually, allowing for more detailed features to be presented.

**The Effectiveness of Iterative Shape Modeling.**    Figure 5 already presents additional examples of our iterative refinement, in addition, Figure 10 illustrates the impact of our global-update procedure, showing how a pronounced axis misalignment is automatically corrected in subsequent refinement steps.

**Ablation Study on Wrapped Library.**    By providing our coding agent with wrapper methods library and its documentation, we observe that the code generated is significantly shorter and without most boilerplate code, thus more context length will be available for holding reasoning content instead of repeated Blender Python code. This is visualized in Figure 11 with a single node generation task.

**Ablation Study on Hierarchical Shape Parsing.**    To study the effectiveness of hierarchical shape parsing procedure, we carry out controlled experiments to compare the performance of our pipeline with and without hierarchical parsing step on a small set of 5 marvel [52] functional prompts. For the base parsing method without hierarchical parsing, the agent is prompted to produce the final node listing directly without decomposing the shape into hierarchical layers first. Specifically, removing hierarchical shape parsing damaged mesh generation quality in all aspect, including VQA pass rate, point cloud distance and intersection over ground truth. The result is shown in Table 4.

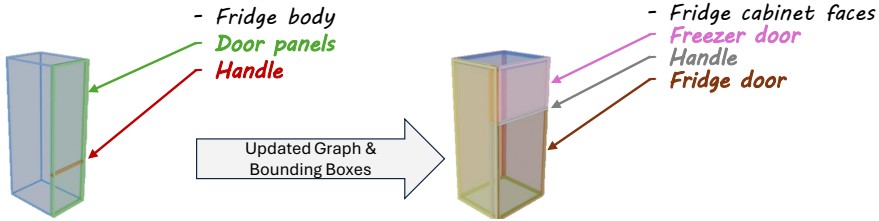

Figure 9: Impact of bootstrapped graph structure and bounding volumes on component alignment. By anchoring the handle node to the positions of both door-panel nodes, the model avoids height discrepancies that arise when panels and handle are generated independently with arbitrary gaps.

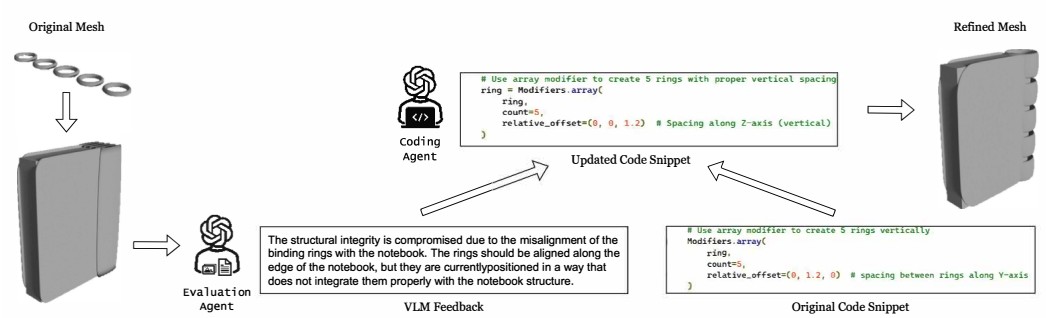

Figure 10: Showcase of global update step solving an axis alignment issue.

Figure 11: Comparison of code generated with and without library method provided.

Table 4: Comparison of ShapeCraft performance with and without hierarchical parsing.

| Method | VQA Pass Rate↑ | Hausdorff Distance↓ | IoGT↑ |
|---|---|---|---|
| No Hierarchical Parsing | 0.48 | 0.564 | 0.297 |
| ShapeCraft | **0.56** | **0.447** | **0.396** |

# D   Additional Qualitative Results

**From components to global mesh.**    Figure 12 illustrates an example shape modeling workflow of ShapeCraft, from iterative updates of individual node shapes, to fitting them into corresponding bounding volumes and obtaining the global raw mesh. Note also the effectiveness of our iterative shape modeling pipeline, updating the Wi-Fi antenna to "slightly angle outward" and fixing the model clipping issue for the buttons.

**CAD modeling.**    Although not designed specifically for CAD modeling, ShapeCraft can accept CAD modeling prompts and model typical CAD shapes either as an entire shape program or as a component geometry, as shown in Figure 13.

**Post-modeling shape editing.**    To showcase the advantage of our shape program representation, we provide an example of prompting agents further based on more shape editing requests based on existing shape program in Figure 14.

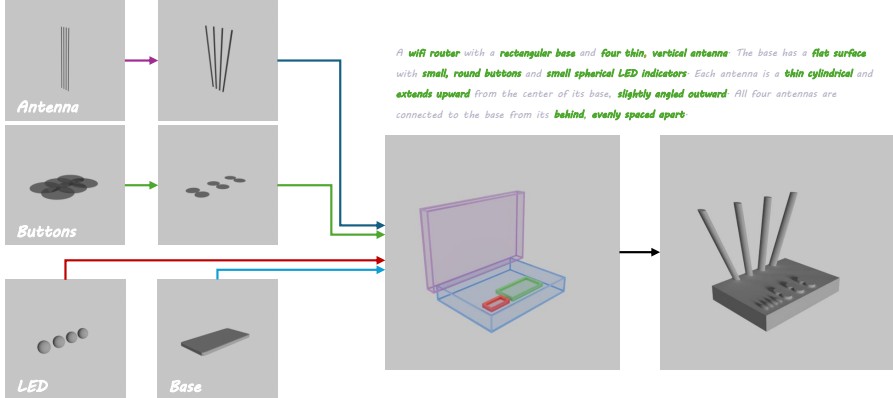

Figure 12: Illustration of the iterative modeling workflow of ShapeCraft using the Wi-Fi router example.

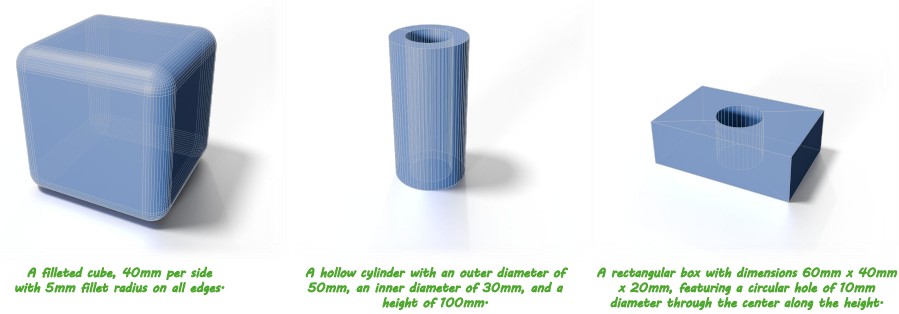

Figure 13: Qualitative CAD Modeling Results: ShapeCraft demonstrates generalizability to CAD modeling tasks despite not being designed specifically for CAD modeling, benefiting from the fact that CAD shapes are relatively sipmler than daily objects but require a higher precision - note that our system does not incorporate accurate measurements into the feedback loop so may produce suboptimal CAD designs.

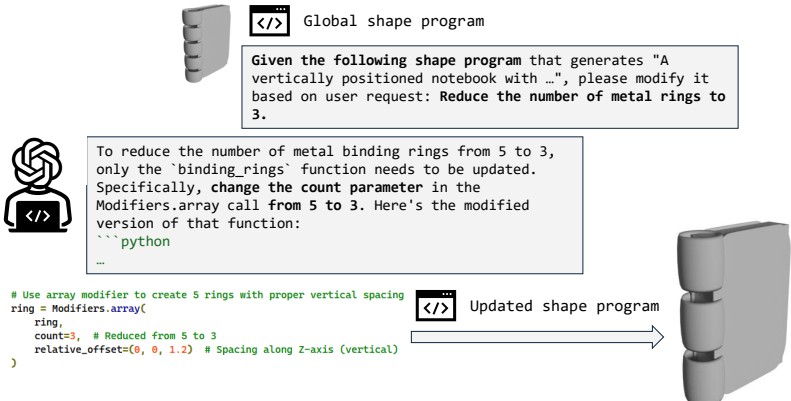

Figure 14: Example of post-modeling shape editing conducted by prompting LLM with existing shape programs directly.

**Integration with native 3D generation methods.** As mentioned in the limitations (Section 5), ShapeCraft struggles in modeling shapes with highly complex topology or organic details, however, this can be mitigated by delegating certain tasks to native 3D generation methods which are more suitable to express these geometric features - in this way, ShapeCraft is still advantageous in terms of

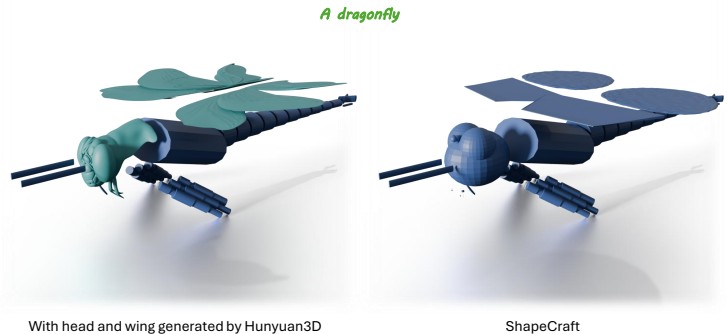

Figure 15: Example of using native 3D generation methods as local shape modeling tool to handle complicate, complex and organic shapes.

scalability due to shape decomposition in the GPS representation. We show in Figure 15 an example where the modeling of a dragonfly's head and wings are done by calling an external API, Hunyuan3D [31], and then fitted to corresponding bounding boxes in the GPS.

**Showcase of ShapeCraft workflow.** As part of our workflow, an example chat history of a shape modeling task for a single node is constructed in Figure 16, where we showcase how error messages are propagated for bug fixes and how visual feedback is applied.

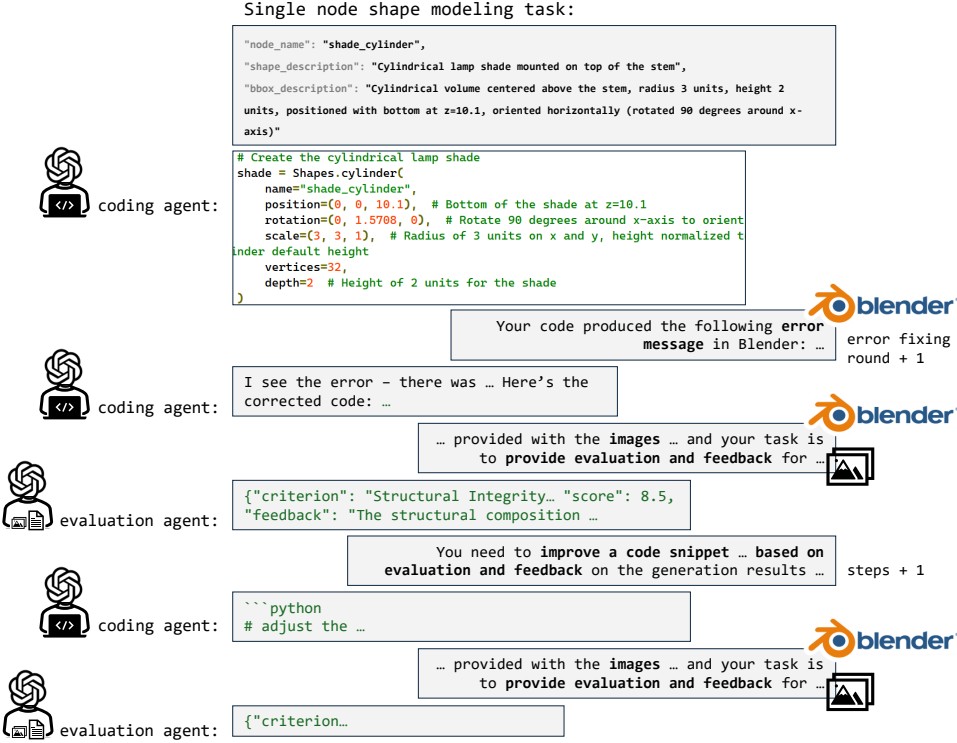

Figure 16: Example chat history for individual shape modeling task.

# E    Prompts Design

We provide a selection of crucial prompts to our pipeline. Specifically, prompt 1 is for shape parsing, prompt 2 and  3 for bootstrapping shape representation. Prompt 4 shows the instruction of generating shape program as bounding boxes based on DAG (an alias for our graph-based representation), and prompt 5 shows the evaluation criteria used in visual feedback for all shape program generation

tasks. For node generation, we use the instruction shown as prompt 6 and provide it with the wrapper method library documentation attached as prompt 7.

Prompt 1: Hierarchical Shape Parsing

```
Given a shape description, decompose the shape into a hierarchical representation
where upper layers are semantic sections and lower down is actual physical
components that can be 3D modeled. In the end, convert the hierarchical
representation in a Directed-Acyclic Graph format with purely leaf nodes,
accompanied with descriptions on their bounding volume. You may use virtual concepts
 to help your group the layers at first, but in the final node representations,
please make sure every node corresponds to a physical component, not concepts or
things that can not be represented by actual 3D models. Make sure you name nodes
properly so that they are suitable for use as Python method names directly. The DAG
should be in JSONL format, where each line represents a node. Return only one
wrapped jsonl code block that contains all the jsonl lines. When there are repeated
or mirrored elements, please group them together in the same node and describe their
 combined bounding volume as well as their shared individual shape - do not
enumerate and create a lot of nodes.

Use the following format:
# First, the hierarchical layers
- root: <section 1 name>, <section 2 name>...
- <section 1 name>: <subsection 1 name>, ...
...
- <subsection name>: <actual physical component name>...
- <section 2...

# Then, the node representations: (note how 'Componment Name' is written as '
component_name' which is suitable for use as Python method names directly)
```jsonl
{"node": <component_name>,
"shape_description": <shape description>,
"bounding_volume": <description on its position, orientation and size, etc.>
}
...
```

Shape: <shape description>
```

Prompt 2: Visual Feedback for Bootstrapping GPS

```
A python script has been generated with respect to the nodes in your DAG
representation that generates bounding boxes for each node, and the generated
bounding boxes has been rendered into an image with different colours. Please try
identifying issues caused by your DAG decomposition (e.g. missing nodes, incorrect
relationships, etc.) and provide feedback on how could the DAG be improved.

# Bounding Box Colours

<color mappings>

# Your Feedback
```

Prompt 3: Updating Shape Representation

```
A python script has been generated with respect to the nodes in your DAG
representation that generates bounding boxes for each node. Some feedback has been
generated by looking at these bounding boxe regarding how to improve the DAG. Please
 provide an updated DAG representation in the same hierarchical layers -> jsonl
nodes format regarding the feedback. Please return the full updated DAG instead of
just the changed parts.

# Feedback

<feedback>

# Your Updated DAG Representation
```

## Prompt 4: Bounding Box Generation (without boilerplate formatting instructions)

```
# Bounding Box Generation Instruction

You will be given a shape description of an object and a Directed Acyclic Graph (DAG
) representation of the object's components and their relationships. Each node in
the DAG represents a component of the object, and is accompanied with descriptions
of their own shape and bounding volume. Your task is to write Python methods for
each node that generates a suitable bounding box based on the bounding volume
descriptions defined in the DAG. Treat each node as a whole and always generate only
 one single bounding box in each node's method. When a node contains repeated
instances, follow bounding volume instructions to generate a single bounding box
that contains all of them. Focus on geometry and ignore other properties like
texture or material in the shape description.

You may use the following wrapper method directly to create a cube as bounding box:
- **cube_bounding_box(name="node_name_bbox", position=(0, 0, 0), scale=(1, 1, 1))**:
 Generates a cube. The tuples are in x-y-z order and z-axis points upward. Parameter
 values are floats. Make sure to use the same name as the node name in the DAG and
suffix it with "_bbox". It returns the object reference of the created cube and make
 sure you return it too.
```

## Prompt 5: Evaluation Criteria (For bounding boxes, the visual quality criterion is removed

```
On a scale of 1 to 10 (significant flaws score 1, generally acceptable score 5, and
perfect examples score 10):
- **Structural Integrity**: Is the generated shape structurally sound? Does it have
any missing or broken parts?
- **Geometric Accuracy**: Does the generated shape accurately represent the
geometric properties of the node? Are the dimensions and proportions correct?
- **Alignment with Description**: Does the generated shape match the description
provided in the DAG?
- **Code Validity**: Does the code work as intended, that you can locate and make
sense between the code and the generated shape?
- **Visual Quality**: How visually appealing is the generated shape? Does it look
realistic and well-formed?
```

## Prompt 6: Node Shape Program Generation

```
You need to write a Python method to create a shape in Blender. Specifically, an
object has been decomposed into a Directed Acyclic Graph (DAG) representation and
you are in charge of writing the code for a specific node in the DAG. The node
represents a component of the object and comes with descriptions of its own shape
and its bounding volume, and you need to create the geometry for that component only
. You will be given the overall DAG representation of the object along with the
shape description of the node you are working on. Note that you don't need to adjust
 positions based on the bounding volumes, since it will be fitted automatically
afterwards. Your code can use a set of wrapper methods to create shapes and apply
modifiers, their documentation is provided below. Focus on geometry and ignore other
 properties like texture or material in the shape description.
```

## Prompt 7: Wrapped Blender Libraries

```python
# Generates a cube.
cube(name, position=(0,0,0), rotation=(0,0,0), scale=(1,1,1))

# Creates a UV-sphere with customizable segments and rings.
sphere(name, position=(0,0,0), rotation=(0,0,0),
        scale=(1,1,1), segments=32, rings=16)

# Adds a cylinder with adjustable vertex count and height.
cylinder(name, position=(0,0,0), rotation=(0,0,0),
          scale=(1,1,1), vertices=32, depth=2)

# Creates a cone with specified base radius, height, and vertex count.
cone(name, position=(0,0,0), rotation=(0,0,0),
      scale=(1,1,1), vertices=32, radius=1, depth=2)

# Adds a plane with adjustable size.
plane(name, position=(0,0,0), rotation=(0,0,0),
       scale=(1,1,1), size=2)

# Creates a 3D Bezier curve. fill_caps closes top/bottom when beveling.
bezier_curve(name, points, bevel_depth=0.0, extrude=0.0,
              fill_caps=False, to_mesh=True)

# Generates a NURBS circle with specified radius and resolution.
circle(name, location=(0,0,0), radius=1.0, segments=32,
        bevel_depth=0.0, extrude=0.0, to_mesh=True)

# Creates straight-line segments. closed links ends; fill_caps closes ends.
polyline(name, points, closed=False, bevel_depth=0.0,
          extrude=0.0, fill_caps=False, to_mesh=True)

# Generates 3D text with LEFT, CENTER, or RIGHT alignment.
text(name, text="Text", location=(0,0,0), size=1.0,
      align='CENTER', extrude=0.0, bevel_depth=0.0, to_mesh=True)

# Constructs a square-based pyramid via vertex & face data.
pyramid(name, position=(0,0,0), rotation=(0,0,0),
         scale=(1,1,1), base_size=2, height=2)

# Creates a capsule by combining hemispheres and a cylinder.
capsule(name, position=(0,0,0), rotation=(0,0,0),
         scale=(1,1,1), radius=1, height=2, segments=32)

# Generates an n-sided prism by configuring a cylinder.
prism(name, position=(0,0,0), rotation=(0,0,0),
       scale=(1,1,1), sides=3, radius=1, height=2)

# Performs INTERSECT/UNION/DIFFERENCE; removes obj_b if remove=True.
Modifiers.boolean(obj_a, obj_b, operation='DIFFERENCE', remove=True)

# Adds subdivision modifier; levels for viewport, render_levels for render.
Modifiers.subdivision(obj, levels=2, render_levels=3)

# Adds bevel modifier; affect=EDGES or VERTICES.
Modifiers.bevel(obj, width=0.1, segments=3, affect='EDGES')

# Duplicates object linearly count times with relative offset.
Modifiers.array(obj, count=5, relative_offset=(1.2, 0, 0))

# Mirrors object across X/Y/Z axes; use_clip prevents crossing plane.
Modifiers.mirror(obj, axis=(True, False, False), use_clip=True)

# Deforms obj along a curve_obj.
Modifiers.curve(obj, curve_obj, deform_axis='POS_X')

# solidify: Adds thickness to a mesh.
Modifiers.solidify(obj, thickness=0.2)

# to_mesh: Applies all modifiers and converts to mesh.
Modifiers.to_mesh(obj)
```

