# OpenReview forum: "ShapeCraft: LLM Agents for Structured, Textured and Interactive 3D Modeling"
_NeurIPS.cc/2025/Conference — NeurIPS 2025 poster_

### Official Review · Reviewer_mrUk · 2025-06-20

**Clarity:** 2
**Significance:** 3
**Originality:** 3
**Rating:** 4
**Confidence:** 4

**Summary:**

This paper presents a method of using LLM to generate 3D shape and texture. The main contribution is proposing a hierarchical structure of the target 3D shape, Graph-based procedural shape representation (GPS) to improve the quality of the generated 3D shapes. Meanwhile, the use of procedural modeling is helpful for generating better results compared with previous methods.

**Questions:**

Followings are some suggestions to improve the expositions of the paper:
- I would recommend to mention the underlying program is based on Blender API. For example, when the GPS representation is introduced.
- The description of the "Hierarchical shape parsing via CoT reasoning" and "Representation bootstrapping." are both very unclear and confusing. I think those function notations are not helping readers to understand what these processes are. I understand that there is a page limit of a neurips submission, and I think this is why I strongly recommend the authors to focus on just shape generation. Therefore, there will be space to describe these processes in more details.
    - For example, how "DecomposeHierarchy" is done? also how "BuildGraph" is done?
- Although it is not ncessary to do a comparison, I am wondering why there is no discussion between the proposed method and BlenderLLM [Du et al. 2024]?

**Ethical Concerns:**

["NO or VERY MINOR ethics concerns only"]

**Final Justification:**

The authors addressed my questions during rebuttal, and I think their latest experiment results are nice to have in the paper.
Therefore, I raise my ratings.

**Limitations:**

It is a bit unclear what is the limitation of the proposed method.
Even though there is a limitation paragraph in Sec. 5, I feel that they are very vague and confusing.
Therefore, I think the limitations are not well discussed, and strongly recommend the authors to provide examples and explanations on this part so that readers can understand the proposed method better.

**Quality:**

3

**Strengths And Weaknesses:**

Strength:
- The proposed method generate nice shape and the use of hierarchical structure is sound.
- The proposed method can also generate texture beside the geometry.

Weakness:
- Meanwhile, I think the paper tends to propose too many content at a time, and therefore fail to describe the method thoroughly in the main paper. Although some details are provided in the supplemental material, it is still makes the paper hard to follow and understand.
- I personally would argue that the paper should focus on the shape generation solely.
- The generated texture quality is not good and not better than previous methods shown in Figure 3. The texture is quite rough and do not match to the generated shape and prompt accurately. This again makes me think this paper should focus on the shape generation.

---

> ### Author Rebuttal · Authors · 2025-07-31
>
> Thank you for your careful reading and valuable feedback. Below please find our clarification regarding your comments.
>
> **W1. Clarity and Scope of Method Presentation**
> We appreciate the reviewer’s feedback and acknowledge that the inclusion of multiple modules (e.g., GPS representation, multi-path modeling, component-aware painting) may impact clarity. In the revision, we will:
> - Clarify the GPS representation and shape modeling with improved notation and explanation.
> - Provide a simplified overview figure with clear annotations of each stage.
> - Relocate implementation details to the appendix and bring key results and analyses from the appendix into the main text.
> We aim to make the system easier to follow without sacrificing technical completeness.
>
> ---
>
> **W2. Texture Quality and Scope of Contribution**
> While shape generation is the primary focus of ShapeCraft, the texture module is included to demonstrate the extensibility of the GPS representation for practical workflows. The diffusion model is invoked as a functional component within the LLM agentic pipeline, conditioned on the GPS structure and executed post shape modeling.
>
> Our component-aware SDS adds node-level descriptions and geometry to guide texturing. We conduct an ablation study of the 10 prompts in [Tab.1](), this design achieves higher alignment with the prompt compared to the original SDS:
>
> | Metric          | Original SDS | Component-Aware SDS |
> | --------------- | ------------ | ------------------- |
> | CLIP Score ↑    | 27.1         | **28.7**            |
> | VQA Pass Rate ↑ | 0.39         | **0.47**            |
>
> In Fig. 3, our method successfully follows texture-related prompts, e.g., generating _“rust and dirt spots, primarily in brown and orange”_ for the air conditioner. In cases with vague or missing texture descriptions, the model falls back to common material priors (e.g., metallic for fridge, leather-like for sofa).
>
> Compared to prior LLM-based shape modeling methods, we are the first to integrate a texture generation module, highlighting a promising direction for future research, even though visual fidelity remains limited.
>
> We will clarify this scope in the revision and emphasize that texture generation is an optional extension of our procedural modeling framework.
>
>
> ---
>
> **Q1. Clarifying Use of Blender API**
> Thank you for the helpful suggestion. We agree that explicitly stating our use of the Blender API when introducing the GPS representation would improve clarity. In the revision, we will update Section 3.1 to clarify that each code snippet $P_i​$ in the GPS graph ultimately corresponds to Blender API calls.
>
> ---
>
> **Q2. Clarification on Hierarchical Parsing and Representation Bootstrapping**
> Thank you for the thoughtful feedback. We acknowledge that the original notation in [Eq.1]() and [Eq.2]() may appear opaque, and we will revise the paper to provide a clearer, more intuitive explanation in the main text.
>
> **Hierarchical Shape Parsing via CoT Reasoning**
> As described in [Eq.1](), `DecomposeHierarchy` refers to the first step of LLM reasoning, where the model breaks down the input shape description into semantic levels of abstraction. We give an example "chair" in [Line 150-151]().
>
> Next, `ParseNodes` converts each leaf in the hierarchy into a **node-level description**, which is then passed to `BuildGraph`, a **deterministic Python procedure** that instantiates the GPS representation (e.g., generates component identifiers, bounding box placeholders, and empty code slots). These are stored in memory and serialized as `gps.json` (see examples in our supplementary material).
>
> **Representation Bootstrapping**
> As shown in [Eq.2]() and illustrated in[ Appendix Fig. 2](), this process refines the initial GPS layout using **visual feedback**:
> 1. ``GenerateBoundingBoxShapeProgram``: The LLM generates an initial bounding box program $G_0$.
> 2. ``ExecuteShapeProgram``: The boxes are rendered into multi-view images $I$.
> 3. ``VlmFeedback`` A VLM is prompted with the rendered images $I_{i-1}$ and asked to evaluate completeness, correctness, and spatial coherence.
> 4. ``GraphUpdate``: The bounding box coder agent revises the previous bounding box program $G_{i-1}$ with the VLM feedback $F_{i-1}$ resulting in updated results $G_i$.
> 5. This loop continues (or early stopping) until a satisfactory version is selected for downstream modeling.
> We will revise Section 3.1 and 3.2 to more clearly explain these two modules in intuitive language, minimizing notation where possible.
>
> ---
>
> **Q3. Comparison with BlenderLLM**
> Thank you for raising this. We have conducted a comparison with BlenderLLM using the same prompts as in [Tab.1](). As shown below, ShapeCraft outperforms BlenderLLM across all metrics, particularly in compile success rate:
>
> | Method     | Hausdorff ↓ | IoGT ↑    | CLIP ↑    | Compile Rate ↑ |
> | ---------- | ----------- | --------- | --------- | -------------- |
> | BlenderLLM | 0.511       | 0.455     | 26.99     | 50%            |
> | ShapeCraft | **0.415**   | **0.471** | **27.27** | **100%**       |
>
> BlenderLLM is tailored for CAD modeling, and its LLM is fine-tuned on instructive prompts and simple shapes. This makes it less effective on open-ended natural language prompts from the MARVEL benchmark. We will include this comparison in the revised version and clarify the distinction in scope and generalizability between the two systems.

---

> > ### Comment · Reviewer_mrUk · 2025-08-03
> >
> > I have read the rebuttal and I appreciate the authors' new experiments that addressed my concerns.
> > I still very curious and concerned about the possible failure cases..
> > Is it possible for authors to explain this a bit further?

---

> > > ### Author Response · Authors · 2025-08-03
> > >
> > > We appreciate your concern on our failure cases. We addressed similar concerns in our response to reviewer C1ui, and to further elaborate, despite the strengths of our GPS-based shape modeling framework and the use of representation bootstrapping, certain scenarios can still present significant challenges.
> > >
> > > **1. Limitations due to Insufficient or Ambiguous Prompts:**
> > >
> > > Our approach relies heavily on the information provided in the textual prompt to generate a reasonable GPS initialization. When the prompt is extremely brief, lacks effective geometric information or is inherently ambiguous (e.g., "a simple speaker" or a creative prompt such as "a sofa that looks like an apple"), the initial parse may fail to decompose the prompt into meaningful nodes. Although subsequent updates and modules like **Representation Bootstrapping** will have the opportunity to correct for a bad GPS initialization, precious conversation rounds are going to be wasted on error correction and the system may produce suboptimal results. In worst cases, the parser agent may experience meaningless and looped updates and cannot produce a rational GPS representation, causing subsequent modeling steps to fail. E.g. for "a simple speaker", the final shape may lack geometric details, or in the worst case being just a cube. For the apple-sofa example, the bounding boxes may not even make sense and the system may only yield a random combination of primitive shapes.
> > >
> > > **2. Limitations with Highly Detailed or Complex Structures:**
> > >
> > > Another failure mode arises when modeling shapes with highly intricate or fine-grained geometric parts that exceed the representational capacity of our current wrapper library. For instance, our library currently struggles to accurately model detailed animal features such as mouths, ears and tails, which often require fine topology and subtle structural nuances. However, our library can also be easily extended to accommodate more expressive abilities, enabling our system to model more types of shapes. In extreme cases, when the prompts exceeds the modeling capabilities of Blender API, our system can also switch to a more suitable shape engine with the method documentation being the only necessary change.
> > >
> > > **3. Potential Solutions and Ongoing Work:**
> > >
> > > To address these challenges, we have experimented with integrating native 3D models (e.g., Hunyuan3D) as add-ons within our library. By invoking such models for generating complex parts, we aim to enhance the capability of our system to handle more sophisticated shapes. Preliminary experimental results demonstrate the feasibility of this approach, suggesting that our framework can potentially accommodate more complicated shapes while preserving structural topology and enabling post-modeling editing.

---

> > > > ### Comment · Reviewer_mrUk · 2025-08-06
> > > >
> > > > Thanks for the detailed explanation.
> > > > I think these discussions are worth included in the revision, and if possible, put corresponding failure cases in the revision.
> > > >
> > > > thank you.

---

> > > > > ### Author Response · Authors · 2025-08-06
> > > > >
> > > > > Thank you for your feedback. We will incorporate these discussions into our revised manuscript. Your comments have been invaluable in helping us strengthen the quality and clarity of our work, and we sincerely appreciate the time and effort you invested in your review.

---

### Official Review · Reviewer_C1ui · 2025-06-28

**Clarity:** 3
**Significance:** 3
**Originality:** 4
**Rating:** 4
**Confidence:** 4

**Summary:**

This paper introduces ShapeCraft, a novel system that leverages Large Language Model (LLM) agents to autonomously generate structured, textured, and interactive 3D models from natural language instructions. The core innovation is the Graph-based Procedural Shape (GPS) representation, which decomposes 3D shapes into hierarchical components represented as nodes in a directed acyclic graph (DAG). Each node contains a natural language description, bounding volume, and code snippet, enabling flexible programmatic updates and post-modeling edits. Experiments demonstrate that ShapeCraft outperforms existing LLM-based and optimization-based methods in geometric accuracy, semantic richness, and editability. The system also supports post-modeling interactions like animation and customization, making it practical for real-world applications.

**Questions:**

Overall, I think this is a nice try of how to combine VLM agent with text-to-3D task. Please check for my concerns as listed below:
(1) The iterative refinement depends on VLM feedback for error correction. What happens if the VLM fails to detect geometric flaws (e.g., intersecting meshes, incorrect proportions)? Is there a risk of error propagation?
(2) The paper claims efficiency gains from parallel sampling, but no concrete runtime comparisons are provided. How does the computational cost scale with shape complexity compared to end-to-end diffusion-based methods?
(3) The BRDF-based texturing is conditioned on rendered images, but does it handle complex material interactions (e.g., transparency, subsurface scattering) or only simple reflectance properties?
(4) The paper shows successful examples but does not analyze where the method fails (e.g., ambiguous prompts, complex topology). A thorough discussion of limitations would strengthen the work.

**Ethical Concerns:**

["NO or VERY MINOR ethics concerns only"]

**Final Justification:**

Thanks for the rebuttal. The authors did address my concerns, and this is a good work. Although the reliance on LLMs may reduce some performance, as demonstrated, there are methods to minimize the impact. Therefore, I will keep my score.

**Limitations:**

Yes.

**Quality:**

3

**Strengths And Weaknesses:**

ShapeCraft represents a notable advancement in text-to-3D generation by combining LLM agents with a structured procedural representation. The strengths and weaknesses are listed as follows:

** Strength
(1) The GPS representation bridges the gap between natural language instructions and procedural 3D modeling, enabling structured, editable outputs compatible with industry tools like Blender. Besides, hierarchical decomposition and bounding volumes simplify spatial reasoning for LLMs, improving accuracy.
(2) Multi-path sampling and VLM feedback loops enhance robustness, allowing the system to explore diverse solutions and correct errors iteratively.
(3) The system supports post-modeling edits and animations, making it valuable for artists and developers. It also demonstrates the superiority over existing methods in quantitative metrics (IoGT, CLIP score) and qualitative results (e.g., fewer artifacts).

** Weakness
(1) The system's performance hinges on the reasoning and coding capabilities of underlying LLMs, which may introduce inconsistencies or failures for complex or ambiguous prompts.
(2) While the GPS representation is flexible, its effectiveness for highly abstract or novel shapes (e.g., fantastical designs) is not thoroughly evaluated. The paper focuses on static objects; dynamic or deformable shapes (e.g., animated characters) are not addressed.
(3) The iterative refinement process, though efficient, may still require significant computational resources for high-quality outputs. The paper lacks detailed analysis of computational costs compared to baseline methods.

---

> ### Author Rebuttal · Authors · 2025-07-31
>
> Thank you for your careful reading and valuable feedback. Below please find our clarification regarding your comments.
>
> **W1. LLM Dependency for complex or ambiguous prompts**
> We thank the reviewer for raising this important point. To mitigate LLM inconsistencies under complex or ambiguous prompts, ShapeCraft integrates several mechanisms:
> 1. **Hierarchical Decomposition + Graph-Based Representation**
> 	We first decompose the input prompt into semantically coherent components using structured chain-of-thought (CoT) reasoning. To further reduce ambiguity, we **augment each node's description with geometry-level details** (e.g., shape, size cues), which improves the LLM’s ability to reason about structure and generate accurate shape programs.
> 	This hierarchical strategy narrows the search space by assigning each component a well-scoped modeling subtask. As discussed in [Lines 131–132](), each node maintains its own component-level search space, enabling the generation of more precise and consistent code. These geometry-aware sub-prompts serve as clear and focused entry points for downstream modeling.
>
> 2. **Representation Bootstrapping with VLM Feedback**  As shown in [Appendix Fig. 2](), initial LLM outputs (e.g., missing the “freezer” component) are visually verified via rendered bounding boxes. Feedback from the VLM guides corrections of hallucinations, omissions, and spatial errors, improving reliability.
>
> 3. **Multi-Path Sampling and Iterative Refinement**
>     Rather than relying on a single output, we explore multiple modelling paths (Sec. 3.2). Each is refined with VLM feedback independently (as shown in [Fig. 5A, Appendix Fig. 3]()), increasing robustness and reducing sensitivity to any single LLM failure.
> Together, these design choices convert complex prompts into manageable sub-tasks, enhance alignment with visual semantics, and significantly improve stability across varied prompt scenarios.
>
> ---
>
> **W2. Effectiveness of GPS representation for abstract, fantastical and dynamic shapes**
> We thank the reviewer for this thoughtful question. ShapeCraft is designed primarily for static, structured shapes, which already present a significant challenge for current LLMs. Our system is most effective when prompts describe geometrically grounded, component-based objects. For highly abstract or fantastical shapes, current LLMs often fail to parse meaningful structure, limiting the reliability of GPS representation. We acknowledge this limitation and plan to explore hybrid approaches (e.g., native 3D model add-on) for such cases in future work.
>
> That said, our Blender wrapper library is modular and extensible, allowing integration with various add-ons or simulation toolkits. This flexibility lays a foundation for handling abstract, fantastical components, given the right parsing operations and prompt design.
>
> For dynamic or articulated shapes, we provide a preliminary demonstration in [Fig. 5(B)]() and in the [Appendix video](), where animations are applied directly to generated components (e.g., opening/closing laptop). This is made possible by our explicit component-based GPS representation, which avoids the need for post-hoc mesh segmentation.
>
> ---
>
> **W3 & Q2. Computational cost**
> We report the average number of API calls and runtime for the methods evaluated in [Tab. 1](). API calls in our setup are categorized into code generation $\mathcal{C}$, feedback $\mathcal{F}$, and question generation $\mathcal{Q}$. We follow this taxonomy to compare against 3D-PREMISE ($\mathcal{C}$+$\mathcal{F}$) and CADCodeVerify ($\mathcal{C}$+$\mathcal{Q}$+$\mathcal{F}$), both evaluated with 3 update steps.
> ShapeCraft consists of two phases: **GPS initialization** and **iterative modeling**. As detailed in [Appendix Sec. A L7–9](), GPS initialization involves 3 API calls for bounding box generation. The iterative modeling phase issues $P\times T$ modeling attempts. Thus, the total API calls for ShapeCraft are: $3+P×T×(\mathcal{C}+\mathcal{F})$. Under a minimal configuration (P=1, T=1), ShapeCraft outperforms previous LLM-based baselines in both quality and computational cost.
>
> We further compare against diffusion-based methods. As shown in [Fig. 3]() and [Fig. 4](), our procedural generation produces more structured and editable meshes, whereas diffusion-based outputs often suffer from marching-cube artifacts and topological noise, limiting their usability in downstream tasks.
>
> In particular, ShapeCraft achieves comparable or better IoGT than MVDream (distill from diffusion model), while being significantly faster. Compared to TRELLIS-text-large (diffusion-based native 3D model trained on 500K shapes using 64 A100 GPUs), ShapeCraft is **training-free** yet approaches its performance. Notably, our evaluation set (Objaverse-MARVEL) is included in TRELLIS's training set, whereas ShapeCraft generalizes without task-specific fine-tuning.
>
> We also demonstrate ShapeCraft’s flexibility in **post-modeling edits** (e.g., animation) in [Fig. 5(B)]() and [Appendix Fig. 6](), highlighting its practicality for interactive applications.
>
> | Method                         | Hausdorff dist.↓ | IoGT↑     | Num of API Calls↓ | Run Time (min)↓ | Train Time (min) ↓|
> | ------------------------------ | ---------------- | --------- | ----------------- | --------------- | ---------------- |
> | 3D-PREMISE (LLM-based)         | 0.527            | 0.385     | 6                 | 2.81            | 0                |
> | CADCodeVertify <br>(LLM-based) | 0.511            | 0.334     | 9                 | 3.06            | 0                |
> | MVDream (diffusion-based)      | **0.411**        | 0.427     | -                 | 32.1            | 0                |
> | TRELLIS (diffusion-based)      | **0.411**        | **0.573** | -                 | **1.05**        | >7200            |
> | **Ours (P=1, T=1)**            | 0.485            | 0.436     | **5**             | **1.62**        | 0                |
> | **ShapeCraft (P=3, T=3)**      | **0.415**        | **0.471** | 21                | 11.68           | 0                |
>
> ---
>
> **Q1. Risk of VLM Failure in Iterative Refinement**
> We acknowledge this limitation. While our VLM-guided bootstrapping significantly improves initial outputs, it is not perfect. To mitigate this:
> - We adopt component-level refinement, so even if one node's error is missed, it does **not propagate globally**.
> - Our multi-path sampling strategy introduces diversity and redundancy, increasing the chance of discovering a correct variant.
> - In practice, VLM feedback successfully catches most common structural issues (e.g., missing parts, overlaps), as shown in [Fig. 5A]() and [Appendix Fig.2,3,4]().
>
> ---
>
> **Q3. Complex material interactions**
> We acknowledge that BRDF alone cannot fully capture translucent or subsurface effects like glass, skin, or wax. However, our texture field design is modular and extendable. By incorporating BTDF models (e.g., adding transmission coefficients and subsurface weight in [Eq.(3)]()), our pipeline can be extended to support complex material types.
>
> ---
>
> **Q4. Failure Case Analysis and Limitations**
> We appreciate the reviewer’s suggestion. We acknowledge two main sources of failure in ShapeCraft:
>
> 1. **Ambiguous or Underspecified Prompts**
>     ShapeCraft relies on prompts containing **explicit geometric cues**. When prompts are vague or lack structural details, the shape parsing may fail, or the upsampled node descriptions may include hallucinated components. For example, prompts like “a magical object with flowing form” cannot be reliably decomposed into actionable modeling steps.
>
> 2. **Complex Topologies**
>     Our current wrapper library is optimized for modeling structured shapes. It lacks native support for fine-grained or highly organic details (e.g., tails, ears, branches). However, we are actively experimenting with extensions, such as integrating Hunyuan3D add-ons, to support more diverse and irregular topologies.
> We will include a detailed discussion of these limitations in the revision, and present qualitative failure cases and possible mitigation strategies.

---

### Official Review · Reviewer_wFFw · 2025-07-02

**Clarity:** 2
**Significance:** 2
**Originality:** 2
**Rating:** 4
**Confidence:** 4

**Summary:**

ShapeCraft proposes a procedural 3D generation pipeline using a multi-agent framework. It presents Graph-based Procedural Shape Representation (GPS) for expressing 3D shapes as programs. The method decomposes objects into components as individual nodes with description, bounding volume and an attached code snippet. When generating an objects, the LLM parses the instruction hierarchically into various components to build the complete object. It further applies bootsrapping using a VLM as a judge to correct for errors in the generated shape. The authors also propose to iteratively sample multiple possible creative paths in parallel to capture the full diversity. To add texture to the generated shape, the method follows score distillation sampling and focuses on it at component level as well as global view. The method is comapred against 4 recent baselines as well as an ablation is conducted to show the need for iterative updates which allows better instruction following by correcting errors.

**Questions:**

* Can the authors compare the average token cost of ShapeCraft against other baselines?
* The method says it uses a pre-defined chain of thought and the authors choose to disable the thinking module. Given the advances in thinking models, is an explicit chain of thought model needed and is the improvement substantial?
* How does the LLM understand the proportion ratios?

**Ethical Concerns:**

["NO or VERY MINOR ethics concerns only"]

**Final Justification:**

Most of my concerns have been adressed by the authors and I think the method will be useful to the community. However, I think the authors should revise the related work to properly cite past work done in this area.

**Limitations:**

Limitations can be expanded such as token cost and generation fidelity compared to generative methods such as Trellis, reliance on SDS for texture

**Quality:**

2

**Strengths And Weaknesses:**

## Strengths
* The paper is mostly well-written and all components are explained.
* The results show good gains in generation quality compared to baselines (both qualitative and quantitative)

## Weaknesses
* Limited novelty-- Decomposition using LLM and graph representation has already been studied quite extensively. The related work also fails to mention many papers doing very similar things- Holodeck, LayoutGPT, FlairGPT, etc. FlairGPT in particular (and many others) decomposes a scene, builds a graph, calls code snippets to build the scene, which is quite similar and hence should be mentioned in related work.
* The iterative refinement based on visual feedback is expensive and also explored before (3D-GPT, SceneCraft: An LLM Agent for Synthesizing 3D Scene as Blender Code [different from this paper although they have the same name], BlenderAlchemy). These references should also be added.
* The texture is generated separately using a SDS based pipeline. I find the author's naming of component-aware SDS a bit gimmicky since it's a simple gathering of node points and there is no real novelty here.
* While the method sees stronger results compared to baselines, the underlying LLM used in these earlier works are weaker. ShapeCraft uses the newly released Qwen3-235B and it becomes very tricky to understand if the performance gains come from the underlying LLM/prompts or the method in question.

---

> ### Author Rebuttal · Authors · 2025-07-31
>
> Thank you for your careful reading and valuable feedback. Below please find our clarification regarding your comments.
>
> **W1. Novelty and Comparison with Scene Generation Works**
>
> **We respectfully disagree with the claim that our GPS representation is similar to prior scene generation approaches** such as Holodeck, LayoutGPT, FlairGPT, or SceneCraft. These methods focus on scene layout construction using LLMs to generate graphs over object bounding boxes and then retrieve pre-existing 3D assets (e.g., from Objaverse). Their graph representations encode coarse inter-object spatial relationships, but are not designed for procedural shape modeling.
>
> In contrast, ShapeCraft focuses on generating novel 3D shapes from scratch. Our Graph-based Procedural Shape (GPS) representation encodes component-level semantics, where each node includes both a bounding volume $B_i$ and an executable modeling program $P_i$. Unlike layout graphs, $P_i$ captures fine-grained geometric construction logic, not just object placement.
>
> We also argue that shape decomposition is inherently more challenging than scene decomposition, as it requires semantic and functional understanding of substructures within a single object, rather than arranging independent entities in space.
>
> We will clarify this distinction in the revised Related Work section under "LLM Agents for Scene Layout" and include a focused discussion of the above methods.
>
> ---
>
> **W2.1 Iterative refinement based on visual feedback is expensive.**
>
> **We respectfully disagree that our refinement strategy is "expensive."** ShapeCraft is designed to be efficient and lightweight:
> 1. We use open-source LLMs/VLMs (e.g., Qwen3-235B), enabling low-cost, local deployment without reliance on GPT-4.
> 2. Our multi-path sampling supports parallelized component modeling, as nodes are independent post-decomposition.
> 3. We employ early stopping based on VLM feedback scores to avoid unnecessary queries.
> 4. Despite being training-free, ShapeCraft outperforms fine-tuned methods like LLaMA-Mesh on both geometry and alignment metrics.
>
> ---
>
> **W2. 2 Comparison with 3D-GPT, SceneCraft and BlenderAlchemy**
> - 3D-GPT builds scenes using the Infinigen function library; it does not support shape modeling or procedural decomposition.
> - SceneCraft **(distinct name from our ShapeCraft)** performs scene-level layout generation using a constraint satisfaction algorithm and retrieves assets.
> - BlenderAlchemy focuses on material optimization through visual feedback loops, rather than structural modeling.
>
> In contrast, ShapeCraft decomposes shape descriptions into fine-grained components and introduces a node-level multi-path sampling strategy. Our VLM feedback loop operates at the per-node level, comparing rendered geometry and component descriptions—not full-scene images—thereby improving both feedback specificity and modeling granularity. We will clarify this discussion in the revised Related Work section.
>
> ---
>
> **W3. Clarification on painting module**
> The diffusion model is invoked as a callable module within our pipeline, it is tightly integrated into the LLM-driven agentic workflow—triggered after shape modeling and conditioned on the GPS structure. Thus, the painting module is not separate, but a downstream step in the procedural generation process.
>
> Our method differs from optimization-based pipelines (e.g., MVDream) that regress entire textures. Instead, we learn a BRDF field over global and local UV coordinates, enabling a physically meaningful and post-editing-friendly texture.
>
> Compared to original SDS, which optimizes the texture globally, our component-aware Score Distillation also introduces node-level conditioning: each component’s geometry and textual description guides rendering and optimization for its corresponding region. This improves semantic alignment, as shown in the following ablation over 10 prompts:
>
> | Metric          | Original SDS | Component-aware Score Distillation |
> | --------------- | ------------ | ---------------------------------- |
> | CLIP Score ↑    | 27.1         | **28.7**                           |
> | VQA Pass Rate ↑ | 0.39         | **0.47**                           |
>
> ---
>
> **W4. Fairness for comparison.**
> To ensure a fair and controlled comparison, we explicitly re-implemented all baselines using the same LLM/VLM setup as ShapeCraft, specifically Qwen3-235B-A22B and Qwen-VL-Max (as stated in [Appendix A Line 5]()). This isolates the performance gain to our GPS representation and multi-path sampling strategy for iterative modeling, not LLM scaling or instruction tuning. We will make this clarification more explicit in the main paper to avoid confusion.
>
> ---
>
> **Q1 & Limitation. Computational resource and token cost.**
> We report the average token cost, runtime and the number of API calls for all methods in [Tab.1](). ShapeCraft incurs higher token usage due to its generation of more detailed, component-wise programs, including multiple code snippets per object.
>
> However, token count alone does not reflect actual computational cost. We also report runtime and API call counts. Despite modest increases in these metrics, ShapeCraft delivers up to 41% IoGT improvement over CADCodeVerify, demonstrating a favorable trade-off. Moreover, under a minimal configuration (P=1, T=1), ShapeCraft achieves both better performance and lower runtime than other LLM-based baselines.
>
> **Limitation: Compared to Trellis**
> Compared to TRELLIS-text-large (diffusion-based native 3D model trained on 500K shapes using 64 A100 GPUs) in [Tab.1](), ShapeCraft is **training-free** yet approaches its performance. Notably, our evaluation set (Objaverse-MARVEL) is included in TRELLIS's training set, whereas ShapeCraft generalizes without task-specific fine-tuning. And ShapeCraft produces procedural generation, which is more structured (in  [Fig. 3]() and [Fig. 4]()) and editable meshes ([Fig. 5(B)]() and [Appendix Fig. 6]()), whereas Trellis outputs often suffer from marching-cube artifacts and topological noise
>
> |                               | Hausdorff dist.↓ | IoGT↑     | Token Cost (K)↓ | Num of API Calls↓ | Run Time (min)↓ | Train Time (min)↓ |
> | ----------------------------- | ---------------- | --------- | --------------- | ----------------- | --------------- | ----------------- |
> | 3D-PREMISE (update 3)        | 0.527            | 0.385     | 14.4            | 6                 | 2.81            | 0                 |
> | CADCodeVertify <br>(update 3) | 0.511            | 0.334     | 16.2            | 9                 | 3.06            | 0                 |
> | TRELLIS (diffusion-based)     | **0.411**        | **0.573** | -               | -                 | **1.05**        | >7200             |
> | **Ours (P=1, T=1)**           | 0.485            | 0.436     | 61.3            | **5**             | **1.62**        | 0                 |
> | **ShapeCraft (P=3, T=3)**     | **0.415**        | **0.471** | 669.1           | 21                | 11.68           | 0                 |
>
>
> ---
>
> **Q2. Compared to LLM with the “thinking” module.**
> As described in [Line 144](), we adopt a hierarchical CoT strategy during shape parsing: the LLM first infers a global semantic root node $H_0$, then recursively decomposes it into component-level nodes $H_i$ based on semantic and structural relationships. This structured CoT constrains the reasoning space and leads to more reliable and interpretable initialization.
> However, we disable verbose or unconstrained CoT (indicate "thinking" module in LLM/VLM) to ensure efficiency and determinism. This decision is based on our observation that free-form CoT often introduces redundant steps or hallucinated geometry operations.
>
> To evaluate our hierarchical CoT strategy, we conducted an ablation using open-ended “thinking” models for prompts in [Tab.1](). As shown below, free-form CoT reasoning fails to maintain spatial consistency across components and often produces invalid results. We report compile rate as the percentage of prompts yielding a valid result in a single run:
>
> | Metrics      | ChatGPT-o3 | ChatGPT-o4-mini-high | Deepseek-R1-0528 | Gemini-2.5-Pro | Ours      |
> | ------------ | ---------- | -------------------- | ---------------- | -------------- | --------- |
> | IoGT↑        | 0.177      | 0.244                | 0.326            | 0.102          | **0.471** |
> | Hausdorff↓   | 0.708      | 0.493                | 0.489            | 0.586          | **0.415** |
> | CLIP↑        | 25.48      | 26.3                 | **29.01**        | 27.31          | 27.27     |
> | Compile Rate | 60%        | 80%                  | 80%              | 60%            | **100%**  |
>
> ---
>
> **Q3. How does the LLM understand the proportion ratios?**
> If the concern relates to how the LLM determines component scale and proportion, we clarify as follows: The LLM infers relative proportions based on its pretrained spatial priors (position embedding) and component descriptions (e.g., “a thin screen above a wide base”). During GraphInit, it generates all bounding boxes in an autoregressive run, so later boxes are conditioned on earlier ones, ensuring internal proportional coherence.
>
> We allow the LLM to freely choose coordinate scales and apply normalization to a unit cube post-hoc to maintain consistency across shapes. Further refinement via VLM feedback ensures spatial accuracy when needed.

---

### Official Review · Reviewer_nYcR · 2025-07-03

**Clarity:** 3
**Significance:** 2
**Originality:** 2
**Rating:** 4
**Confidence:** 3

**Summary:**

This paper introduces ShapeCraft, an LLM agent system that generates structured, textured, and interactive 3D models from text. Its core innovation is a Graph-based Procedural Shape (GPS) representation, where an LLM decomposes an object into a graph of components, each defined by a description, bounding box, and a procedural code snippet. An agentic workflow with Vision-Language Model (VLM) feedback iteratively generates and refines the geometry for each part. The resulting models are well-aligned with the prompt, structured, and editable, a significant advantage over existing methods. Experiments show ShapeCraft outperforms a range of text-to-3D approaches.

**Questions:**

- During the initial GraphInit phase, how are the bounding boxes (Bi) for each component determined? Does the LLM generate numerical coordinates and dimensions, and if so, how does it reason about the spatial coherence of the overall layout before the VLM-based bootstrapping step?
- Could you elaborate on the design of the Blender Wrapper Library? For instance, what level of abstraction does it provide? How significant is the role of this library in constraining the search space and simplifying the task for the LLM?
- Regarding the multi-path sampling in Algorithm 1, what were the typical values used for the number of paths (P) and steps (T) in your experiments? How much diversity did you observe across the generated paths? Did they explore genuinely different modeling strategies or tend to converge to similar final shapes?

**Ethical Concerns:**

["NO or VERY MINOR ethics concerns only"]

**Final Justification:**

Most of my earlier concerns have been addressed through the rebuttal and additional clarifications, and the work demonstrates clear novelty and robustness. However, uncertainties remain regarding its scalability to extremely complex topologies, so I will maintain the borderline accept score.

**Limitations:**

yes

**Quality:**

2

**Strengths And Weaknesses:**

Paper strengths:

- The Graph-based Procedural Shape (GPS) representation design allows for the generation of structured and editable assets.
- The experimental evaluation is strong. ShapeCraft shows superiority over a wide range of state-of-the-art methods in both raw mesh and textured mesh generation.

Paper weaknesses:

- Lack of deeper analysis: The system's success is heavily reliant on the capabilities of powerful, and likely proprietary, LLMs and VLMs. The paper would benefit from a more detailed analysis of failure modes related to these models (e.g., API hallucination, misinterpretation of geometric instructions). Furthermore, more quantitative ablation studies of each design are needed, such as the Blender Wrapper Library and different hyperparameters of multi-path sampling.
- Scalability to complex topologies: The examples presented (e.g., furniture) are primarily compositions of relatively simple geometric primitives. It is unclear how well the approach would scale to generating objects with highly organic or intricate topologies, such as animals, human characters, or complex mechanical parts, where procedural generation from primitives might be insufficient.
- Complexity and computational cost: The overall pipeline is quite complex, involving multiple feedback loops and parallel sampling paths. A more thorough discussion of the computational cost (in terms of runtime and, more importantly, the number of API calls to LLMs/VLMs) would be valuable; understanding its cost relative to other agentic systems is important.

---

> ### Author Rebuttal · Authors · 2025-07-30
>
> Thank you for your careful reading and valuable feedback. Below please find our clarification regarding your comments.
>
> **W1.1 Concern on LLM/VLM Dependency and Failure Modes**
> To address potential issues such as LLM hallucination and geometric misinterpretation, we designed two core mechanisms:
> 1. **Representation Bootstrapping** As shown in [Appendix Fig.2](), initial LLM parsing misses components (e.g., "freezer" in the fridge example). Our bootstrapping mechanism (Sec. 3.1) introduces a VLM-based visual verification loop: the LLM’s output is rendered as bounding-box geometry and compared to the textual input. Feedback from the VLM guides the correction of omissions, hallucinations, and spatial inconsistencies.
> 2. **Multi-Path Sampling with Iterative Updates**  Instead of relying on a single deterministic program, we adopt multi-path sampling (Sec. 3.2), which explores diverse shape programs in parallel. Each path undergoes iterative refinement with VLM feedback to correct execution errors or misalignments (as shown in [Fig. 5(A) and Appendix Fig.3]()). This redundancy (lowers the risk of not having any compiled shape programs in the end) improves robustness **against LLM/VLM inaccuracies and API hallucinations**. As a result, ShapeCraft achieves a significantly higher compile success rate compared to baseline methods and even strong CoT-based reasoning models.
>
> **Failure Modes**
> Even equipped with above functions, ShapeCraft may fail when provided with ambiguous or underspecified prompts. In such cases, shape parsing may produce hallucinated or incomplete node descriptions, making downstream modeling unreliable. For instance, prompts like _“an amazing chair with complicated decoration”_ cannot be decomposed into actionable components.
>
> ---
>
> **W1.2 & Q2: Blender Wrapper Library and Its Quantitative Ablation Study**
> Qualitative ablation study of wrapper library can be found in [Appendix Fig. 5](). We further conducted a quantitative ablation study over 10 prompts subset below:
>
> | Method                          | Avg. Code Length ↓  | Median Code Length ↓ | Hausdorff ↓ | CLIP Score ↑ |
> | ------------------------------- | ------------------- | -------------------- | ----------- | ------------ |
> | w/o Wrapper Library             | 189.57              | 111.00               | 0.424       | 27.1         |
> | ShapeCraft (w/ Wrapper Library) | **168.40 (-11.2%)** | **77.50 (-30.2%)**   | **0.447**   | **27.3**     |
>
> Our Blender Wrapper Library is designed to reduce the complexity of the API space exposed to the LLM. And results show that Blender Wrapper Library provide better shape results and produce more concise program. It abstracts frequent modeling operations—such as primitive creation, Boolean ops, and modifiers—into a curated set of consistent, high-level commands. Instead of raw Blender Python calls (e.g., `bpy.ops.mesh.primitive_cube_add` with multiple parameters), we use simplified wrappers like `cube(location, rotation, scale)`. These abstractions align better with LLM priors and reduce trial-and-error programming during code generation. Detailed API can be found in [Appendix prompt 6]().
>
> ---
>
> **W1.3 & Q3: Ablation on Multi-Path Sampling**
> Thank you for pointing this out. We provide more quantitative ablations and clarifications below.
> In our experiments, we used **P=3 and T=3** for shape modeling, and **P=1, T=3** for bounding box generation mentioned in [Appendix Sec. A, L7–9](). These settings strike a balance between performance and efficiency.
>
> To analyze diversity, we visualize multiple sampled paths in [Appendix Fig. 3](). Empirically, we observe that: 1) For simple shapes, paths tend to converge to similar modeling strategies due to low ambiguity. 2) For complex or ambiguous prompts, different paths often explore genuinely distinct decomposition strategies and shape programs, especially under higher temperature settings. In general, we observe 2–3 unique strategies across 3 paths for complex cases, demonstrating the system's capacity to explore modeling alternatives.
>
> We also conduct quantitative ablation with varying (P, T) to evaluate the benefit of multi-path sampling. Results below:
>
> | Sampling Setting          | Hausdorff ↓ | IoGT ↑    | CLIP Score ↑ | Run Time (min)↓ |
> | ------------------------- | ----------- | --------- | ------------ | --------------- |
> | P=1, T=1                  | 0.485       | 0.436     | 25.75        | **1.62**            |
> | P=1, T=3                  | 0.494       | 0.492     | 26.20        | 3.90            |
> | P=3, T=1                  | 0.444       | **0.535** | 25.90        | 3.71            |
> | P=3, T=5                  | **0.360**   | 0.431     | 26.39        | 18.04           |
> | **ShapeCraft (P=3, T=3)** | 0.415       | 0.471     | **27.27**    | 11.68           |
>
> These results show that multi-path sampling significantly improves both geometry quality (IoGT, Hausdorff) and prompt alignment (CLIP), confirming its effectiveness.
>
> ---
>
> **W2. Scalability to complex topologies.**
> We appreciate the reviewer’s concern. While many examples focus on furniture, ShapeCraft is not restricted to simple primitives. As shown in [Fig. 4](), our _Stichter Banjo_ example includes curved surfaces and interlocking parts, illustrating the system’s ability to model non-trivial geometries.
>
> Importantly, our Blender Wrapper Library is extensible, supporting advanced operations like subdivision, bevel, and boolean. Since our agent generates tool-based programs, ShapeCraft can scale to any backend modeling API, given appropriate documentation.
>
> Moreover, ShapeCraft’s hierarchical decomposition and modular program structure naturally support complex assemblies such as CAD parts or articulated components. In revisions, we will include qualitative results on CAD-style shapes.
>
> ---
>
> **W3. Complexity and computational cost.**
> Our method scales well with increasing shape complexity and number of parsed components, due to the parallel execution of per-node modeling tasks. The use of our Blender Wrapper Library also reduces the overall program length by abstracting frequent operations, which helps conserve context window compared to native Blender APIs. This is supported by code length comparisons in [Appendix Fig. 5]() and discussed in [W1.2 & Q2]().
>
> We also report the number of API calls and runtime for the methods in [Tab.1](). API calls are mainly related to code generation $\mathcal{C}$, feedback $\mathcal{F}$ and question generation $\mathcal{Q}$ (only used in CADCodeVertify). ShapeCraft consists of GPS representation initialization and iterative shape modeling. GPS representation initialization needs 3 API calls with one trajectory of three updates for bounding box generation mentioned in [Appendix Sec. A, L7–9](). The number of Iterative shape modeling depends on the $P$ and $T$. Under a minimal configuration (P=1, T=1), it outperforms previous LLM-based methods both in quality and computational cost:
>
> | Method                        | Hausdorff dist.↓ | IoGT↑     | Num. of API Calls↓                                     | Run Time (min) ↓ |
> | ----------------------------- | ---------------- | --------- | ------------------------------------------------------ | ---------------- |
> | 3D-PREMISE (update 3)         | 0.527            | 0.385     | 6: $3\times (\mathcal{C}+\mathcal{F})$                 | 2.81             |
> | CADCodeVertify <br>(update 3) | 0.511            | 0.334     | 9: $3\times (\mathcal{C}+\mathcal{Q}+\mathcal{F})$     | 3.06             |
> | **Ours (P=1, T=1)**           | 0.485            | 0.436     | **5**:  $3+P\times T \times (\mathcal{C}+\mathcal{F})$ | **1.62**         |
> | **ShapeCraft (P=3, T=3)**     | **0.415**        | **0.471** | 21: $3+P\times T \times (\mathcal{C}+\mathcal{F})$     | 11.68            |
>
>   We will include this analysis in the revised paper to clarify the trade-offs between program complexity and execution efficiency.
>
> ---
>
> **Q1. Bounding Box Generation and Spatial Coherence in GraphInit**
> Thank you for the thoughtful question. During the GraphInit phase, bounding boxes $B_i$ are generated by the LLM through an upsampled node description $x_i$ that supply relative spatial relationships. The LLM uses a wrapper method in [Appendix Prompt 4]():
>
> `cube_bounding_box(name="node_name_bbox", position=(x, y, z), scale=(sx, sy, sz))`
>
> Although no visual feedback is available at this stage, LLMs exhibit basic spatial reasoning due to their pretraining. We do not impose fixed value ranges, allowing the model to freely choose a coordinate scale; normalization is applied afterward to ensure consistent layout.
>
> Notably, the bounding box agent generates all  $B_i$ in a single autoregressive run. This implicitly conditions each box on previous ones (i.e., models  $P(B_i|B_{<i})$), helping maintain global spatial coherence even before visual grounding.
>
> Subsequently, the bootstrapping phase [Eq. 2]() integrates VLM feedback to further refine placements. This two-stage design—coarse layout via LLM, fine adjustment via vision—strikes a balance between reasoning efficiency and geometric fidelity.

---

> > ### Comment · Reviewer_nYcR · 2025-08-05
> >
> > Thank you for the detailed rebuttal and additional experiments. The clarifications on bounding box generation, the blender wrapper library, and multi-path sampling address some of my earlier questions. However, my main concerns remain regarding scalability to highly complex topologies and the heavy reliance on proprietary LLM/VLM capabilities.

---

> > > ### Author Response · Authors · 2025-08-05
> > >
> > > Thank you for your feedback. We're glad to see that most of your concerns have been addressed. Below, we provide further clarification on the remaining points:
> > >
> > > 1. **Highly Complex Topology**:
> > >
> > > Due to the NeurIPS rebuttal constraints this year, we are unable to include visualizations. However, we can offer further analysis and discussion of representative cases.
> > >
> > > For shapes with complex internal topology - such as engine blocks with intricate internal channels - contemporary 3D generative models typically struggle. They often produce approximate outer forms without meaningful internal structures and frequently fail on non-watertight inputs. ShapeCraft addresses this by integrating agentic planning, which allows the decomposition of modeling tasks and the delegation of subtasks to independent generators. This structured approach enables better handling of topological complexity from a higher-level perspective.
> > >
> > > From the above example, a simple parse we get contains specific modeling tasks for engine body, mounting points, oil channels and coolant channels, which allows these parts to be optimized independently and later combined together to yield a clear topology. In addition, our library can be easily extended to incorporate generative models to further improve visual quality of each components while also maintaining a meaningful overall structure.
> > >
> > > For multi-part topologies (e.g., a spider with eight legs), ShapeCraft’s hierarchical parsing and modular decomposition allow it to scale effectively with the number of components, demonstrating robustness and adaptability in such scenarios.
> > >
> > > 2. **Use of Proprietary LLMs/VLMs**:
> > >
> > > We would like to clarify that ShapeCraft does *not* rely on proprietary LLMs or VLMs. Our controlled experiments and ablation studies use open-source or open-weights models, such as the Qwen series. During development, we also experimented with a broader range of models - including Gemma 3, DeepSeek R1/V3, and proprietary ones like GPT and Gemini - but found that while stronger models offer slight performance gains, the core improvements stem from our pipeline design rather than model choice.
> > >
> > > Furthermore, ShapeCraft consistently outperforms baseline methods under the same LLM/VLM configurations, further demonstrating that its advantages are architectural and algorithmic rather than dependent on proprietary models.

---

### Note · Authors · 2025-08-11

We sincerely thank the reviewers and the Area Chair for their thorough evaluation and constructive dialogue. We're grateful that our detailed rebuttals and the ensuing discussions successfully addressed most concerns, as reflected in the reviewers’ acknowledgements and updated scores.

For the AC’s final consideration, we wish to emphasize two key clarifications that emerged from the discussions:

1. **Architectural Innovation vs. Model Dependency:**

    A primary concern was the reliance on powerful LLMs. We reiterate that ShapeCraft's advantages are **architectural, not model-dependent**. Our core innovation lies in the Graph-based Procedural Shape (GPS) representation and the agentic workflow. To ensure a fair comparison, all baselines were re-implemented and evaluated using the **same open-source / open-weights LLMs** (Qwen series), proving that our performance gains stem from our novel framework design.


2. **Scalability and Handling Complexity:**

    Scalability for complex topologies is a fundamental challenge in the field. ShapeCraft addresses this through **Hierarchical Decomposition** that breaks down complexity into simpler **component-level sub-tasks**. This procedural approach offers a more structured and extensible path toward complex modeling than monolithic generation. Our ShapeCraft achieves performance that is quantitatively close to TRELLIS, a native 3D model that relies on large-scale datasets and incurs heavy training costs. And ShapeCraft dominates existing training-free methods (e.g. CADCodeVerify and 3D-PREMISE) in both quantitative and qualitative results. As illustrated in Figure 1, our method can model **fine-grained structures with high fidelity**, such as the individual keys on a computer keyboard. As discussed, our system is designed to be **extensible**, allowing for the integration of more advanced back-end tools or generative add-ons to handle intricate organic shapes (e.g., animals and characters).

The reviewers’ feedback has been invaluable. We are fully committed to incorporating the full scope of our discussions—particularly the detailed failure case analysis, clarifications on computational cost breakdowns and more qualitative visualizations—into the final manuscript to provide a more robust presentation of our work.

Thank you for your time and consideration.

---

### Decision · Program_Chairs · 2025-09-17

**Decision:**

Accept (poster)

**Comment:**

The paper received four reviews, all initially rated as "Borderline Accept" (4). Following a thorough rebuttal and discussion period, the reviewers maintained their scores, but their justifications clearly shifted to a more positive consensus, acknowledging that the authors had addressed most of their significant concerns.

After going through the paper, the reviews and the discussions, we have accepted the paper.